# Revisiting Ensembling in One-Shot Federated Learning

**Youssef Allouah**[1]    **Akash Dhasade**[1*]   **Rachid Guerraoui**[1]    **Nirupam Gupta**[2]

**Anne-Marie Kermarrec**[1]    **Rafael Pinot**[3]    **Rafael Pires**[1]    **Rishi Sharma**[1]

[1]EPFL    [2]University of Copenhagen
[3]Sorbonne Université and Université Paris Cité, CNRS, LPSM

## Abstract

Federated learning (FL) is an appealing approach to training machine learning models without sharing raw data. However, standard FL algorithms are iterative and thus induce a significant communication cost. One-shot federated learning (OFL) trades the iterative exchange of models between clients and the server with a single round of communication, thereby saving substantially on communication costs. Not surprisingly, OFL exhibits a performance gap in terms of accuracy with respect to FL, especially under high data heterogeneity. We introduce FENS, a novel federated ensembling scheme that approaches the accuracy of FL with the communication efficiency of OFL. Learning in FENS proceeds in two phases: first, clients train models locally and send them to the server, similar to OFL; second, clients collaboratively train a lightweight prediction aggregator model using FL. We showcase the effectiveness of FENS through exhaustive experiments spanning several datasets and heterogeneity levels. In the particular case of heterogeneously distributed CIFAR-10 dataset, FENS achieves up to a $26.9\%$ higher accuracy over state-of-the-art (SOTA) OFL, being only $3.1\%$ lower than FL. At the same time, FENS incurs at most $4.3\times$ more communication than OFL, whereas FL is at least $10.9\times$ more communication-intensive than FENS.

## 1 Introduction

FL is a widely adopted distributed machine learning (ML) approach, enabling clients to *collaboratively train* a common model over their collective data without sharing raw data with a central server [27]. Clients in FL engage in iterative parameter exchanges with the server over several communication rounds to train a model. While providing high accuracy, this process incurs substantial communication cost [19]. One-shot federated learning (OFL) [11] has been introduced to address the communication challenges in FL by reducing the exchange of models to a single round. Not surprisingly, this came with a loss of accuracy with respect to FL.

Typical OFL methods execute local training at the clients up to completion and form an ensemble of locally trained models at the server [7, 10, 46, 11]. The ensemble is distilled into a single model, through means of either auxiliary public dataset [10, 11] or synthetic data generated at the server [7, 14, 46]. While these OFL methods address communication challenges by reducing model exchanges to a single round, they often achieve lower accuracy compared to iterative FL. This is especially true when data distribution across clients is highly heterogeneous as OFL methods typically

---

*Corresponding author <akash.dhasade@epfl.ch>

38th Conference on Neural Information Processing Systems (NeurIPS 2024).

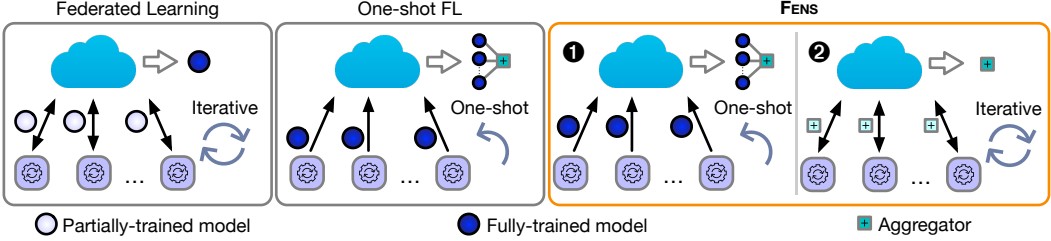

Figure 1: FENS in comparison to iterative and one-shot federated learning.

rely on simple prediction aggregation schemes such as averaging [11, 46], weighted averaging [7, 10] or voting [8].

We introduce FENS, a hybrid of OFL and standard FL. FENS aims to approach both the accuracy of iterative FL as well as the communication cost of OFL. Learning in FENS proceeds in two phases. In the first phase, similar to OFL, clients upload their locally-trained models to the server. Instead of using the traditional OFL aggregation, FENS employs a second phase of FL: the server constructs an ensemble with a prediction *aggregator model* stacked on top of the locally trained models. This advanced aggregation function is then trained by the clients in a lightweight FL training phase. The overall learning procedure is illustrated in Figure 1, alongside iterative and one-shot FL.

## 1.1 Our Contributions

We show for the first time, to the best of our knowledge, that a shallow neural network for the aggregator model suffices to satisfactorily bridge the gap between OFL and FL. Leveraging a shallow aggregator model enables two major benefits: first, it induces significantly lower communication cost in the iterative phase, and second, the iterative refinement of this aggregator model significantly improves accuracy over existing OFL methods. By utilizing elements from both OFL and FL in this novel ensembling scheme, FENS achieves the best of both worlds: accuracy of FL and communication efficiency of OFL.

Through extensive evaluations on several benchmark datasets (CIFAR-100, CIFAR-10, SVHN, and AG-News) across different heterogeneity levels, we demonstrate the efficacy of FENS in achieving FL-like accuracy at OFL-like communication cost. We extend our empirical evaluations to the FLamby benchmark [32], a realistic cross-silo FL dataset for healthcare applications. Our results show that in heterogeneous settings where even iterative FL algorithms struggle, FENS remains a strong competitor. We then conduct an extensive study of different aggregator models and highlight the accuracy vs. communication trade-off. Lastly, we show that FENS maintains high accuracy even with a comparable memory footprint.

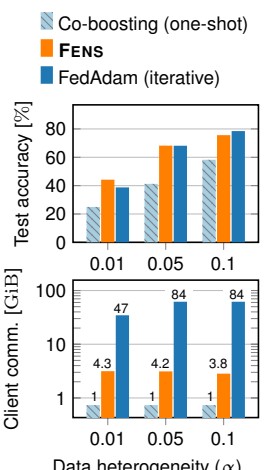

Bar labels indicate the normalized comm. cost w.r.t. OFL

Figure 2: Test accuracy and communication cost of OFL, FENS and FL on CIFAR-10 dataset under high data heterogeneity.

To showcase FENS's performance, we compare its accuracy and communication costs against Co-Boosting [7], a state-of-the-art OFL method, and FEDADAM [33], a popular iterative FL algorithm, as shown in Figure 2. These evaluations are performed on the CIFAR-10 dataset with 20 clients across three heterogeneity levels: $\alpha = 0.01$ (very high), $\alpha = 0.05$ (high), and $\alpha = 0.1$ (moderate). Co-Boosting exhibits an accuracy gap of $13.7 - 26.9\%$ compared to FEDADAM. FENS closes this accuracy gap, being only $0 - 3.1\%$ lower than FEDADAM. To achieve this, FENS incurs only $3.8 - 4.3\times$ more communication than Co-Boosting whereas FEDADAM is $10.9 - 22.1\times$ more expensive than FENS.

## 1.2 Related Work

**One-shot Federated Learning.** Guha *et al.* [11] introduced one-shot FL, which limits communication to a single round. They proposed two main methods: *(i)* heuristic selection for final ensemble clients, and *(ii)* knowledge distillation (KD) for ensemble aggregation into a single model at the server using an auxiliary dataset. Subsequent methods based on KD [10, 22] require large, publicly available

datasets similar to local client data for good performance, which are often difficult to obtain [50]. To address this, synthetic data generation using generative adversarial networks (GAN)s has been proposed [7, 46]. The SOTA Co-Boosting algorithm [7] iteratively generates and refines synthetic data and the ensemble model. In FEDCVAE-ENS [14], clients train variational autoencoders (VAEs) locally and upload decoders to the server, which generates synthetic samples for classifier training. FEDOV [8] trains an open-set classifier at each client to predict "unknown" classes, with the server ensembling these models and using open-set voting for label prediction. Other OFL approaches either do not fully consider data heterogeneity [22, 37], or face difficulties under high data heterogeneity [48].

Another line of research in OFL focuses on aggregating fully trained client model parameters [42, 45]. PFNM [45] matches neurons across client models for fully-connected networks, while FEDMA [42] extends this to convolutional neural networks (CNNs) and LSTMs. However, the performance of these methods drops with more complex models. Few theoretical works exist, such as [18], which analyze global model loss for overparameterized ReLU networks. Despite the advances, OFL still exhibits accuracy gap with iterative FL. We show that FENS narrows this accuracy gap while preserving communication efficiency.

**Ensembles in Federated Learning.** Ensembles have been previously studied in FL for a variety of different goals. FEDDF [24] performs robust model fusion of client ensembles to support model heterogeneity. The FEDBE algorithm [5] uses Bayesian Model Ensemble to aggregate parameters in each global round, improving over traditional parameter averaging. Hamer *et al.* propose FEDBOOST [12] that constructs the ensemble using simple weighted averaging and analyze its optimality for density estimation tasks. However, these works are designed for standard FL and rely on substantial iterative communication. In the decentralized edge setting, [38] show that collaborative inference via neighbor averaging can achieve higher accuracy over local inference alone. However, they assume a setting where clients can exchange query data during inference and consider only IID data replicated on all edge devices. The idea of learning an aggregator model closely resembles late fusion techniques in multimodal deep learning [25]. The key difference is that FENS focuses on fusing single modality models trained on heterogeneous data under the communication constraints of federated settings.

## 2 Description of FENS

We consider a classification task represented by a pair of input and output spaces $\mathcal{X}$ and $\mathcal{Y}$, respectively. The system comprises $M$ clients, represented by $[M] = \{1, \ldots, M\}$ and a central server. Each client $i$ holds a local dataset $\mathcal{D}_i \subset \mathcal{X} \times \mathcal{Y}$. For a model $h_\theta : \mathcal{X} \to \mathcal{Z}$ parameterized by $\theta \in \Theta \subseteq \mathbb{R}^d$, each data point $(x, y) \in \mathcal{X} \times \mathcal{Y}$ incurs a loss of $\ell(h_\theta(x), y)$ where $\ell : \mathcal{Z} \times \mathcal{Y} \to \mathbb{R}$. Denoting by $\mathcal{D} := \bigcup_{i \in [M]} \mathcal{D}_i$ the union of all local datasets, the objective is to solve the empirical risk minimization (ERM) problem: $\min_{\theta \in \Theta} \frac{1}{|\mathcal{D}|} \sum_{(x,y) \in \mathcal{D}} \ell(h_\theta(x), y)$.

### 2.1 Federated Learning (FL) and One-shot FL (OFL)

FL algorithms, such as FedAvg [27], are iterative methods that enable the clients to solve the above ERM problem, without having to share their local data. In each iteration $t$, the server broadcasts the current model parameter $\theta_t$ to a subset of clients $S_t \subseteq [M]$. Each client $i \in S_t$ updates the parameter locally over its respective dataset $\mathcal{D}_i$ using an optimization method, typically stochastic gradient descent (SGD). Clients send back to the server their locally updated model parameters $\{\theta_t^{(i)}, i \in S_t\}$. Lastly, the server updates the global model parameter to $\theta_{t+1} := \frac{1}{|S_t|} \sum_{i \in S_t} \theta_t^{(i)}$.

In One-shot Federated Learning (OFL), the iterative exchanges in FL are replaced with a *one-shot* communication of local models. Specifically, each client $i$ seeks a model $\theta^{(i)}$ that approximately solves the ERM problem on their local data: $\min_{\theta \in \Theta} \frac{1}{|\mathcal{D}_i|} \sum_{(x,y) \in \mathcal{D}_i} \ell(h_\theta(x), y)$, and sends $\theta^{(i)}$ to the server. Upon receiving the local parameter $\theta^{(i)}$, corresponding to parametric model $\pi_i = h_{\theta^{(i)}}$, the server builds an ensemble model of the form $\pi(x) = \sum_{i \in [M]} w_i \pi_i(x)$. This ensemble model is then distilled into a single global model at the server using either a public dataset or synthetic data (generated by the server). Existing OFL algorithms choose weights $w_1, \ldots, w_M$ in three different ways: *(i)* uniformly at random [46], *(ii)* based on local label distributions [10], and *(iii)* dynamically adjusted based on generated synthetic data [7].

## 2.2 FENS

In FENS, the server builds the ensemble model using a generic aggregator $f_\lambda : \mathcal{Z}^M \to \mathcal{Z}$, parameterized by $\lambda \in \Lambda \subset \mathbb{R}^q$ to obtain a global model $\pi : \mathcal{X} \to \mathcal{Z}$ defined to be

$$\pi(x) := f_\lambda(\pi_1(x), \dots, \pi_M(x)). \tag{1}$$

In case of standard aggregators such as weighted averaging, $q = M$ and $\lambda \in (w \in \mathbb{R}_+^M \big| \sum_{i=1}^M w_i = 1)$, and $f_\lambda(\pi_1(x), \dots, \pi_M(x)) := \sum_{i \in [M]} \lambda_i \pi_i(x)$. In general, $f_\lambda$ can be a non-linear trainable model such as a neural network. The overall learning procedure in FENS comprises two phases:

1. **Local training and one-shot communication:** Each client $i$ does local training to compute $\theta^{(i)}$, identical to OFL, and sends it to the server.

2. **Iterative aggregator training:** Upon receiving the local parameters $\theta^{(i)}$, the server reconstructs $\pi_i := h_{\theta^{(i)}}$, and obtains $\widehat{\lambda}$ that approximately solves the following ERM problem:

$$\min_{\lambda \in \Lambda} \quad \frac{1}{|\mathcal{D}|} \sum_{(x,y) \in \mathcal{D}} \ell\left(f_\lambda(\pi_1(x), \dots, \pi_M(x)), y\right). \tag{2}$$

The above ERM problem is solved using an iterative FL scheme (described above). For doing so, the server transmits the set of local models $\{\pi_1, \dots, \pi_M\}$ to all the clients. The final model is given by $\pi(x) := f_{\widehat{\lambda}}(\pi_1(x), \dots, \pi_M(x))$.

When solving for (2) using iterative FL, only the aggregator parameters are transferred between the server and the clients. As we show through experiments, in the subsequent section, training an aggregator model is much simpler than training the local models $\pi_i$, and a shallow neural network suffices for $f_\lambda$. Algorithms 1 and 2 (Appendix C) provide the pseudo for FENS.

**Connection with stacked generalization.** The use of a trainable aggregator corresponds to an instance of stacked generalization [44] in the centralized ensemble literature, wherein the aggregation function is regarded as *level* 1 generalizer, while the clients' models are regarded as *level* 0 generalizers. It has been shown that level 1 generalizer serves the role of correcting the biases of level 0 generalizers, thereby improving the overall learning performance of the ensemble [44]. While stacked generalization has been primarily studied in centralized settings, through FENS we show that this scheme can be efficiently extended to an FL setting.

## 3 Experiments

We split our evaluation into the following sections: *(i)* FENS vs OFL in Section 3.2; *(ii)* FENS vs FL and analysis of when FENS can match FL in Section 3.3; *(iii)* FENS on real-world cross-silo FLamby benchmark [32] in Section 3.4; *(iv)* FENS on language dataset in Section 3.5; *(v)* dissecting components of FENS in Section 3.6; and *(vi)* enhancing FENS efficiency in Section 3.7.

### 3.1 Experimental setup

**Datasets.** We consider three standard vision datasets with varying level of difficulty, including SVHN [30], CIFAR-10 [30] and CIFAR-100 [30], commonly used in several OFL works [7, 10, 46] as well as one language dataset AG-News [47]. Vision experiments involve 20 clients, except in the scalability study, where client numbers vary; and AG-News uses 10 clients. The original training splits of these datasets are partitioned across clients using the Dirichlet distribution $\mathtt{Dir}_{20}(\alpha)$, in line with previous works [7, 10, 14]. The parameter $\alpha$ determines the degree of heterogeneity, with lower values leading to more heterogeneous distributions (see Appendix A, Figure 9). For our experiments involving the realistic healthcare FLamby benchmark, we experiment with 3 datasets: Fed-Camelyon16, Fed-Heart-Disease, and Fed-ISIC2019. Table 5 (Appendix A) presents an overview of the selected tasks. The datasets consist of a natural non independent and identically distributed (non-IID) partitioning across clients. In FENS, each client performs local training using 90% of their local training data while reserving 10% for the iterative aggregator training. We observed that by splitting the datasets, we achieve better performance than reusing the training data for aggregator training. For fairness, OFL and FL baselines run with each client using 100% of their dataset for

local training. The testing set of each dataset is split (50-50%) for validation and testing. We use the validation set to tune hyperparameters and always report the accuracy on the testing split.

**One-shot FL baselines.** We compare FENS against 6 one-shot baselines: *(i)* one-round FE-DAVG [27]; *(ii)* FEDENS [11], the first one-shot method constituting an ensemble with uniform weights; *(iii)* FEDKD [10], based on auxiliary dataset; *(iv)* one-shot version of FED-ET [6]; *(v)* the data-free FEDCVAE-ENS [14]; and *(vi)* Co-Boosting [7], based on synthetic data generation. We use the best-reported hyperparameters in each work for the respective datasets wherever applicable or tune them. Appendix B.1 provides additional details regarding the one-shot baselines.

**Iterative FL baselines.** For comparison with FL, we consider 6 algorithms: *(i)* FEDAVG [27]; *(ii)* FEDPROX [23]; *(iii)* FEDNOVA [43]; *(iv)* SCAFFOLD [20]; *(v)* FEDYOGI [33]; and *(vi)* FEDADAM [33]. We tune learning rates for each algorithm. In addition to these baselines, we implement gradient compression with FEDAVG STC, following the sparsification and quantization schemes of STC [29]. In particular, we set the quantization precision to 16-bit and sparsity level to $50\%$, to reduce the communication cost of FEDAVG by $4\times$ and keep the remaining setup to the same as above baselines. For the FLamby benchmark experiments, we use the reported hyperparameters which were obtained after extensive tuning, except with one difference. The authors purposefully restricted the number of rounds to be approximately the same as the number of epochs required to train on pooled data (see [32]). Since this might not reflect true FL performance, we rerun all FL strategies to convergence using the reported tuned parameters. Precisely, we run up to $10\times$ more communication rounds than evaluated in the FLamby benchmark. We include more details on FL baselines in Appendix B.2.

**FENS.**[2] For the CIFAR-10 and CIFAR-100 datasets, each client conducts local training for $500$ epochs utilizing SGD as the local optimizer with an initial learning rate of $0.0025$. For the SVHN and AG-News datasets, local training extends to $50$ and $20$ epochs respectively with a learning rate of $0.01$. The learning rate is decayed using Cosine Annealing across all datasets. For the FLamby benchmark experiments, each client in FENS performs local training with the same hyperparameters as the client local baselines of FLamby. We experiment with two aggregator models, a 2-layer perceptron with ReLU activations and another that learns per-client per-class weights. We train the aggregator model using the FEDADAM algorithm where the learning rate is separately tuned for each dataset (Table 8, Appendix B.3). To reduce the communication costs corresponding to the ensemble download, we employ post-training model quantization at the server from `FP32` to `INT8`. Appendix B.3 provides more details on FENS.

**Configurations.** In line with related work [10, 24, 35], we use ResNet-8 [13] as the client local model for our vision tasks and fine-tune DistilBert [34] for our language task. Our FLamby experiments use the same models as defined in the benchmark for each task (see Table 5, Appendix A). We report the average results across at least 3 random seeds. For iterative FL baselines, the communication cost corresponds to the round in which the best accuracy is achieved.

## 3.2 FENS vs OFL

To assess FENS's efficacy, we experiment in non-IID settings, varying $\alpha \in \{0.01, 0.05, 0.1\}$, and present results across datasets and baselines in Table 1. Our observations reveal challenges for one-shot methods under high heterogeneity, with the optimal baseline differing across settings. FEDAVG with one round exhibits the poorest performance. While FEDCVAE-ENS maintains consistent accuracy across various heterogeneity levels and datasets, it struggles particularly with CIFAR-10 and CIFAR-100, indicating challenges in learning effective local decoder models. Regarding distillation-based methods, FEDKD and Co-Boosting demonstrate similar performance on SVHN and CIFAR-10. However, FEDKD outperforms Co-Boosting on CIFAR-100, facilitated by the auxiliary public dataset for KD while Co-Boosting arduously generates its synthetic transfer dataset. FED-ET improves over FEDENS and is also competitive to FEDKD. Notably, FENS consistently outperforms the best baseline in each scenario, except for SVHN at $\alpha = 0.01$, where FEDCVAE-ENS excels. FENS achieves significant accuracy gains, surpassing the best baseline by $11.4 - 26.9\%$ on CIFAR-10 and $8.7 - 15.4\%$ on CIFAR-100, attributed to its advanced aggregator model.

We chart the client communication costs incurred by all algorithms in Figure 3. The clients in FENS expend $3.6 - 4.3\times$ more than one-shot algorithms owing to the ensemble download and iterative

---

[2]Source code available at: `https://github.com/sacs-epfl/fens`.

Table 1: FENS vs one-shot FL for various heterogeneity levels across datasets. The highest achieved accuracy is presented as bold and the top-performing baseline is underlined. The rightmost column presents the performance difference between FENS and the top-performing baseline.

| Method | $\alpha$ | FEDAVG | FEDENS | FEDKD | FED-ET | FEDCVAE | Co-Boosting | FENS | $\Delta$ |
|---|---|---|---|---|---|---|---|---|---|
| CF-100 | 0.01 | $04.22_{\pm0.33}$ | $16.59_{\pm2.07}$ | $\underline{28.98_{\pm4.55}}$ | $20.37_{\pm1.53}$ | $07.84_{\pm0.98}$ | $14.76_{\pm2.14}$ | $\mathbf{44.46_{\pm0.31}}$ | $+15.4$ |
| | 0.05 | $05.53_{\pm0.13}$ | $20.56_{\pm3.51}$ | $\underline{39.01_{\pm1.11}}$ | $33.20_{\pm1.34}$ | $07.90_{\pm0.83}$ | $20.28_{\pm1.94}$ | $\mathbf{49.70_{\pm0.86}}$ | $+10.6$ |
| | 0.1 | $06.04_{\pm0.92}$ | $27.41_{\pm2.71}$ | $\underline{42.38_{\pm0.78}}$ | $38.53_{\pm1.04}$ | $08.05_{\pm0.69}$ | $25.50_{\pm0.65}$ | $\mathbf{51.11_{\pm0.37}}$ | $+8.7$ |
| CF-10 | 0.01 | $10.35_{\pm0.29}$ | $15.66_{\pm6.11}$ | $18.59_{\pm2.92}$ | $16.94_{\pm7.88}$ | $\underline{29.98_{\pm0.88}}$ | $24.97_{\pm4.72}$ | $\mathbf{44.20_{\pm3.29}}$ | $+14.2$ |
| | 0.05 | $13.56_{\pm3.33}$ | $39.56_{\pm6.33}$ | $38.84_{\pm6.03}$ | $37.51_{\pm2.87}$ | $\underline{29.40_{\pm2.53}}$ | $41.25_{\pm5.88}$ | $\mathbf{68.22_{\pm4.19}}$ | $+26.9$ |
| | 0.1 | $17.38_{\pm0.22}$ | $48.40_{\pm9.01}$ | $\underline{64.14_{\pm5.17}}$ | $47.06_{\pm2.31}$ | $32.10_{\pm1.79}$ | $58.24_{\pm3.54}$ | $\mathbf{75.61_{\pm1.85}}$ | $+11.4$ |
| SVHN | 0.01 | $11.35_{\pm7.05}$ | $20.31_{\pm3.49}$ | $23.62_{\pm10.1}$ | $12.63_{\pm6.23}$ | $\mathbf{69.71_{\pm0.21}}$ | $25.32_{\pm8.94}$ | $57.35_{\pm12.6}$ | $-12.3$ |
| | 0.05 | $12.85_{\pm5.34}$ | $38.91_{\pm7.28}$ | $37.41_{\pm9.62}$ | $41.14_{\pm6.66}$ | $\underline{70.63_{\pm1.59}}$ | $46.36_{\pm3.29}$ | $\mathbf{76.76_{\pm2.98}}$ | $+6.1$ |
| | 0.1 | $27.80_{\pm9.83}$ | $51.99_{\pm7.85}$ | $61.38_{\pm3.90}$ | $58.91_{\pm2.81}$ | $\underline{72.38_{\pm0.28}}$ | $59.19_{\pm7.06}$ | $\mathbf{83.64_{\pm0.75}}$ | $+11.2$ |

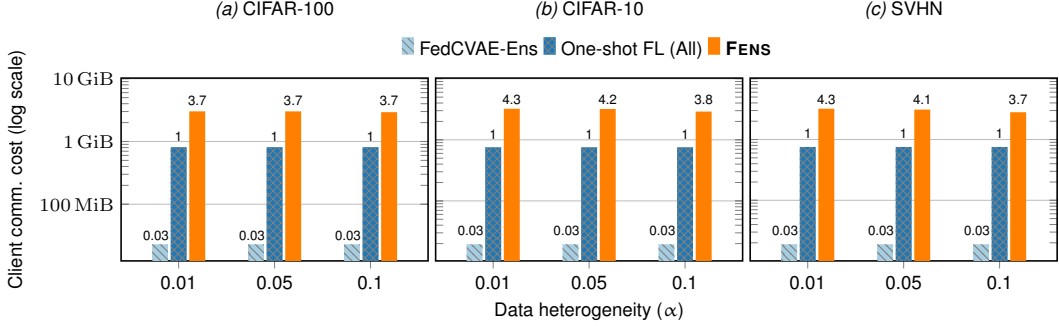

*(a) CIFAR-100*  *(b) CIFAR-10*  *(c) SVHN*

FedCVAE-Ens   One-shot FL (All)   **FENS**

Client comm. cost (log scale)

Data heterogeneity ($\alpha$)

Bar labels indicate the normalized communication cost w.r.t. OFL

Figure 3: Total communication cost of FENS against OFL baselines. The clients in FENS expend roughly $3.7 - 4.3\times$ more than OFL in communication costs.

aggregator training. While these costs are greater than OFL, they are significantly lower than the costs incurred by iterative FL baselines as shown in Section 3.3. FEDCVAE-ENS has the lowest cost since the clients only upload the decoder component of their VAE model to the server. However, it also suffers from a significant performance gap with respect to other OFL baselines and FENS on the CIFAR-10 and CIFAR-100 datasets.

**Varying number of clients.** We also assess the performance of different baselines by varying the number of clients. We consider the CIFAR-10 dataset with $\alpha = 0.1$ and present the results in Table 2. FENS achieves the best accuracy surpassing the best-performing baseline FEDKD by $5.9 - 11.4\%$ in accuracy points. This again demonstrates the benefits of utilizing an advanced ensemble model with a trainable aggregator function.

Table 2: FENS vs one-shot FL on CIFAR-10 with varying number of clients.

| $M$ | FEDAVG | FEDENS | FEDKD | FED-ET | FEDCVAE | Co-Boosting | FENS | $\Delta$ |
|---|---|---|---|---|---|---|---|---|
| 5 | $32.00_{\pm0.69}$ | $62.70_{\pm8.45}$ | $\underline{71.74_{\pm4.11}}$ | $47.83_{\pm2.71}$ | $27.73_{\pm1.50}$ | $60.86_{\pm4.33}$ | $\mathbf{77.70_{\pm2.62}}$ | $+5.9$ |
| 10 | $24.00_{\pm5.68}$ | $46.80_{\pm1.28}$ | $\underline{66.85_{\pm5.18}}$ | $49.22_{\pm4.81}$ | $30.00_{\pm2.21}$ | $54.10_{\pm4.19}$ | $\mathbf{76.44_{\pm2.64}}$ | $+9.5$ |
| 20 | $17.38_{\pm0.22}$ | $48.40_{\pm9.01}$ | $\underline{64.14_{\pm5.17}}$ | $47.06_{\pm2.31}$ | $32.10_{\pm1.79}$ | $58.24_{\pm3.54}$ | $\mathbf{75.61_{\pm1.85}}$ | $+11.4$ |
| 50 | $18.57_{\pm8.64}$ | $49.89_{\pm3.82}$ | $\underline{59.26_{\pm2.90}}$ | $51.59_{\pm2.31}$ | $34.59_{\pm1.26}$ | $51.38_{\pm4.10}$ | $\mathbf{70.53_{\pm1.10}}$ | $+11.2$ |

### 3.3 FENS vs Iterative FL

We now compare the accuracy and communication cost of FENS against iterative FL baselines. After our extensive evaluation of all 6 iterative FL baselines (Table 10, Appendix D) on the CIFAR-10 dataset across various heterogeneity levels, we find that FEDADAM and FEDYOGI consistently perform the best. Hence, we show FEDADAM in our remaining evaluations. Figure 4 presents the results, showing FEDADAM as the representative of FL, FEDAVG STC as the representative of gradient compression, FEDKD as the representative of OFL, and FENS. Moreover, we show two versions of FEDKD, one additional with multi-round support to match the communication cost

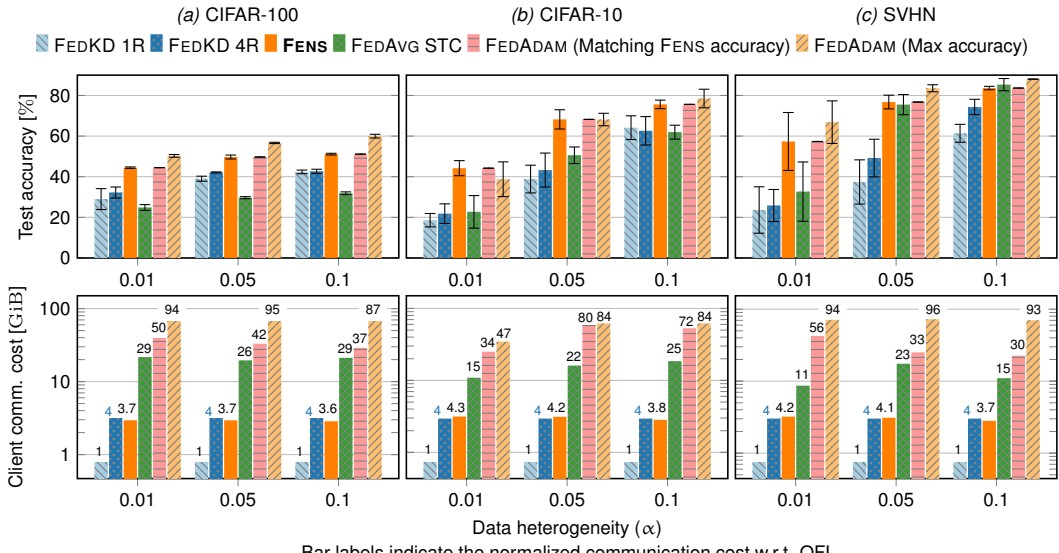

Figure 4: **FENS against iterative FL.** The R indicates the number of global rounds, signifying the multi-round version of the OFL baseline. FENS achieves accuracy properties of iterative FL (FEDADAM) with a modest increase in communication cost compared to OFL (FEDKD). Numerical accuracy results are included in Table 11 (Appendix D).

of FENS (details in Appendix B.2). We also show two versions of FEDADAM, one achieving the accuracy of FENS and the other with its maximum accuracy to facilitate effective comparison of communication costs.

We observe that FENS with its iteratively trained aggregator significantly closes the accuracy gap between OFL (FEDKD) and FL (FEDADAM) across all datasets and heterogeneity levels. Remarkably, the boost achieved is sufficient to match FEDADAM's accuracy at $\alpha = \{0.01, 0.05\}$ on the CIFAR-10 dataset. This comes at only a modest increase in communication costs which are $\approx 4\times$ that of OFL across all cases. We observe that FEDADAM incurs $30 - 80\times$ more communication than OFL to reach the same accuracy as FENS. Even adding multi-round support to FEDKD only marginally improves its performance. While the best accuracy achieved by FEDADAM still remains higher, it also comes at significant communication costs of $47 - 96\times$ that of OFL. Furthermore, we observe that communication compression (FEDAVG STC) fails to preserve the accuracy of FL under high heterogeneity. Thus FENS achieves the best accuracy vs. communication trade-off, demonstrating accuracy properties of iterative FL while retaining communication efficiency of OFL.

### 3.3.1 When can FENS match iterative FL?

In this section, we aim to understand when FENS can match iterative FL. The performance of ensembles depends upon *(i)* the quality of local models; and *(ii)* data heterogeneity. The quality of local models in turn depends on the amount of local data held by the clients. As local models improve at generalizing locally, the overall performance of the ensemble is enhanced. In contrast, FL struggles to generate a good global model when the local datasets of clients significantly differ. Thus more volume of data does not analogously benefit FL due to high data heterogeneity. However, as heterogeneity reduces, FL excels and benefits significantly from collaborative updates. This suggests that FENS under *sufficiently large local datasets* and *high heterogeneity* can match iterative FL's performance. We confirm this intuition through the following experiments on the SVHN dataset.

**Setup.** We study the performance of FL and FENS by progressively increasing the volume of data held by the clients. To this end, we consider the classification task on the SVHN dataset due to the availability of an extended training set of $604\,388$ samples, *i.e.*, $\approx 10\times$ bigger than the default SVHN dataset. We then experiment with fractions ranging from 10 to 100% of the total training set. Each client locally utilizes $90\%$ for one-shot local model training and reserves $10\%$ for iterative aggregator training, similar to previous sections. We then compare FENS with FEDAVG (FL baseline) on three

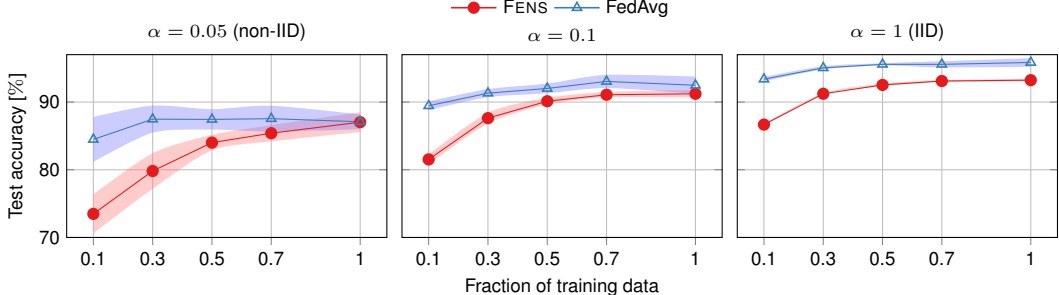

Figure 5: Accuracy of FENS for increasing dataset size. Performance of FENS rapidly increases as the data volume increases. At high data heterogeneity, FENS matches iterative FL's accuracy.

levels of heterogeneity: $\alpha = \{0.05, 0.1, 1\}$, varying from highly non-IID to IID. We tune the learning rate for FEDAVG (details in Appendix B.2) and keep the remaining setup as in previous experiments.

**Results.** Figure 5 shows the results and confirms our prior insight behind the effective performance of FENS. Specifically, we observe that the growing volume of training data benefits FENS much more than FEDAVG. When the data distribution is homogeneous ($\alpha = 1$), the performance of FENS improves faster than FEDAVG, but still remains behind. On the other hand, under high heterogeneity ($\alpha = 0.01$), FENS quickly catches up with the performance of FEDAVG, matching the same accuracy when using the full training set. We conclude that under regimes of high heterogeneity and sufficiently large local datasets, FENS presents a practical alternative to communication expensive iterative FL.

### 3.4 Performance on real-world datasets

In this section, we evaluate the performance of FENS on the real-world cross-silo FLamby benchmark [32]. Specifically, we present 5 iterative baselines and 2 one-shot baselines along with the client local baselines (Figure 6). For the one-shot FEDAVG and FEDPROX OFL baselines, we additionally tune the number of local updates. FEDKD is infeasible in these settings since it requires a public dataset for distillation, unavailable in the medical setting. FEDCVAE-ENS and Co-Boosting are also infeasible due to the difficulty in learning good decoder models or synthetic dataset generators for medical input data, a conclusion supported by their poor performance on the comparatively simpler CIFAR-100 task (Table 1). Figure 6 shows the results with the first row comparing FENS against iterative FL algorithms and the second row against one-shot FL and the client local baselines.

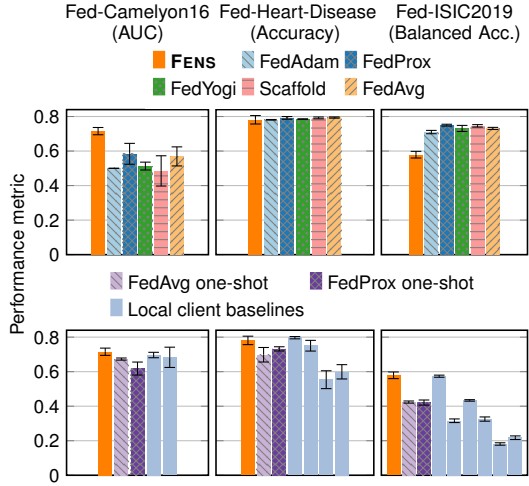

Figure 6: **FENS in FLamby.** FENS is on par with iterative FL (row-1), except when local models are weak (Fed-ISIC2019) while remaining superior in the one-shot setting (row-2). Numerical results included in Tables 12 to 17 (Appendix D).

When comparing to iterative FL, we observe that FENS is on par for the Fed-Heart-Disease dataset and performs better for Fed-Camelyon16. The iterative FL performance is affected by high heterogeneity [32] where the deterioration is more significant for Fed-Camelyon16, which learns on large breast slides ($10000 \times 2048$) than for Fed-Heart-Disease, which learns on tabular data. In such scenarios of heterogeneity, FENS can harness diverse local classifiers through the aggregator model to attain good performance. On the Fed-ISIC2019 dataset, however, FENS does not reach the accuracy of iterative FL. Clients in Fed-ISIC2019 exhibit high variance in local data amounts and model performance, with the largest client having 12k samples and the smallest only 225 (Table 5, Appendix A). We thus speculate that the Fed-ISIC2019 dataset falls within the low local training fraction regime depicted in Figure 5, exhibiting a larger

accuracy gap compared to iterative FL. However, we note that FENS achieves superior performance over one-shot FEDAVG and one-shot FEDPROX, while performing at least as well as the best client local baseline across all datasets. Overall, these observations for FL algorithms have spurred new interest in developing a better understanding of performance on heterogeneous cross-silo datasets [32]. We show that FENS remains a strong competitor in such settings.

### 3.5 Performance on language dataset

We now study the performance of FENS on the AG-News dataset, comparing it against top-performing baselines FEDADAM and FEDKD in the iterative and one-shot categories, respectively. Figure 7 shows the results: at $\alpha = 0.1$, FEDKD achieves $71.5\%$ accuracy, leaving a gap to FEDADAM at $82.3\%$. FENS effectively closes this gap, reaching $78.8\%$. As heterogeneity reduces at $\alpha = 0.3$, all algorithms achieve higher accuracies. FENS improves upon FEDKD from $79.3\%$ to $84.5\%$ while FEDADAM achieves $88.3\%$. Thus we observe consistent results on the language task as our vision benchmarks.

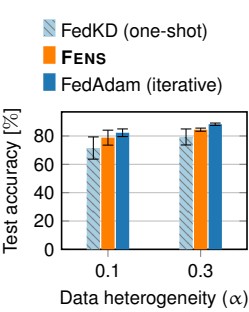

Figure 7: FENS on the AG-News dataset.

### 3.6 Dissecting FENS

We extensively evaluate various aggregation functions (details in Appendix B.4) on the CIFAR-10 dataset across diverse heterogeneity levels. In particular, we assess static aggregation rules including averaging and weighted averaging, parametric aggregator models including a linear model, and a shallow neural network. We also evaluate an advanced version of voting [2] which involves computing competency matrices to reach a collective decision. In addition, we evaluate the Mixture-of-Experts (MoE) aggregation rule [36] where only the gating function is trained via federation. Figure 8 illustrates the accuracy, communication costs, and breakdown for all aggregations. Trained aggregator models outperform static aggregations, incurring additional costs for ensemble download and iterative training. The NN aggregator emerges as the top performer, offering the best accuracy vs. communication trade-off. Notably, the iterative training cost of the NN aggregator model for several rounds is lower than the OFL phase itself. Regarding accuracy, only MoE outperforms NN at $\alpha = 0.01$, where extreme heterogeneity induces expert specialization, while the trained gating network accurately predicts the right expert. However, MoE's performance declines as heterogeneity decreases while its communication costs remain higher due to the size of the gating network.

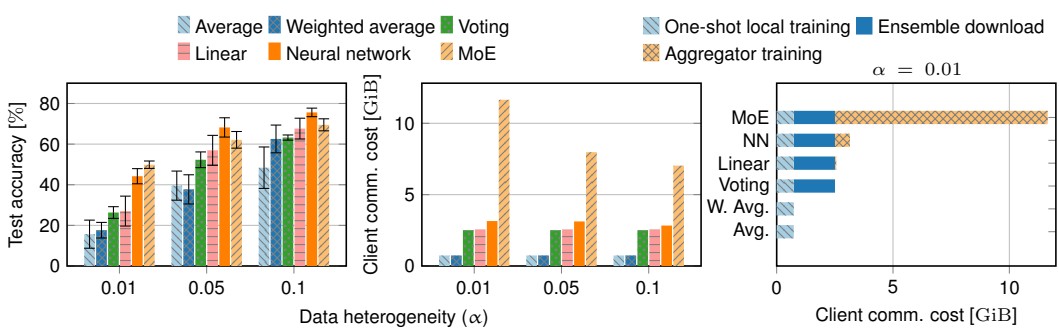

Figure 8: Accuracy of different aggregation functions on the CIFAR-10 dataset. NN offers the best accuracy vs. communication trade-off, with its iterative training taking up only a fraction of the total cost. Numerical accuracy values are included in Table 9 (Appendix D).

### 3.7 Enhancing FENS efficiency

The FENS global model comprises the aggregator model stacked atop the ensemble of client local models. Although FENS achieves strong accuracy, the ensemble model can be computationally and memory intensive. We used FP32 to INT8 quantization in our previous experiments which reduces the memory costs by $4\times$ (Appendix B.3). In this section, we explore two additional approaches to reduce FENS's overheads.

**What if we distill FENS into a single model?** To enable efficient inference, we can distill the FENS global model into a single model at the server using KD once the training is completed. Specifically, we distill the FENS ensemble model comprising 20 ResNet-8 client models and the shallow aggregator neural network into a single ResNet-8 model. Table 3 presents the results on the CIFAR-10 dataset for $\alpha = \{0.01, 0.05, 0.1\}$ distilled using CIFAR-100 as the auxiliary dataset. We observe a slight accuracy drop arising from the process of distillation, which is standard behavior [39]. While we distill using the standard distillation algorithm [15], we note that this accuracy gap can be further reduced through the use of more advanced distillation methods [16, 49].

Table 3: Accuracy of FENS after distillation on the CIFAR-10 dataset.

| Algorithm | $\alpha = 0.01$ | $\alpha = 0.05$ | $\alpha = 0.1$ |
|---|---|---|---|
| FENS | $44.20_{\pm 3.29}$ | $68.22_{\pm 4.19}$ | $75.61_{\pm 1.85}$ |
| FENS distilled | $43.81_{\pm 2.58}$ | $65.56_{\pm 3.25}$ | $71.59_{\pm 1.21}$ |

Table 4: FENS vs FEDADAM under similar memory footprint on CIFAR-10. DS stands for downsized.

| Algorithm | $\alpha = 0.01$ | $\alpha = 0.05$ | $\alpha = 0.1$ | Memory (MiB) |
|---|---|---|---|---|
| FENS | $43.32_{\pm 3.92}$ | $66.99_{\pm 4.01}$ | $74.14_{\pm 1.63}$ | 377.03 |
| FENS-DS | $43.08_{\pm 3.41}$ | $64.59_{\pm 3.74}$ | $72.43_{\pm 2.16}$ | 17.95 |
| FEDADAM | $39.32_{\pm 7.85}$ | $68.74_{\pm 2.76}$ | $78.73_{\pm 3.55}$ | 18.85 |
| FEDADAM-DS | $29.69_{\pm 3.62}$ | $63.77_{\pm 0.23}$ | $72.25_{\pm 3.22}$ | 0.88 |

**What if we match the memory footprint of FENS to FEDADAM?** While the above approach enables efficient inference, we still need to mitigate training time costs. We now consider a downsized (DS) version of ResNet-8, where the width of a few layers is reduced so that the total size of the FENS downsized (FENS-DS) model approximately matches the size of the single model in FEDADAM. Table 4 presents the results on the CIFAR-10 dataset for various heterogeneity levels. Note that no quantization is considered for FENS and FENS-DS, hence the contrast in values with Table 3. Under a comparable memory footprint, FENS-DS remains competitive with the original FEDADAM, with only a slight drop in accuracy compared to FENS. On the other hand, using the downsized model as the global model in FEDADAM-DS results in a significant accuracy drop (from 39.32% to 29.69%) under high data heterogeneity ($\alpha = 0.01$). Thus, the memory overhead of FENS can be alleviated while retaining its communication benefits without too much impact on accuracy.

## 4   Discussion and Conclusion

**Limitations.** One limitation is the memory required on client devices to store the ensemble model for aggregator training. We explored quantization and downsizing to mitigate this issue. Future work could investigate aggregator models that do not require access to all client models in the ensemble. This memory issue is only present during training; after training, FENS can be distilled into a single global model on the server, enabling efficient inference as shown in Section 3.7. Another limitation is the increased vulnerability to attacks during iterative aggregator training, unlike OFL, which limits the attack surface to one round. However, this only affects the aggregator model, since the client local models are still uploaded in one shot. Privacy can be further enhanced in FENS through techniques such as differential privacy [9] or trusted execution environments [28]. Specifically, clients can use differentially private SGD [1] for local training, providing a differentially private local model for the ensemble, while the aggregator training could leverage a differentially private FL algorithm [31].

**Benefits.** In addition to low communication costs and good accuracy, FENS provides three important advantages. First, it supports model heterogeneity, allowing different model architectures across federated clients [21]. Second, FENS enables rapid client unlearning [4], towards the goal of the *right to be forgotten* in GDPR [26]. In FENS, unlearning a client can be achieved by simply re-executing the lightweight aggregator training by excluding the requested clients' model from the ensemble. This is more efficient than traditional FL, where disincorporating knowledge from a single global model can be costly. Lastly, if a small server-side dataset is available, such as a proxy dataset for bootstrapping FL [3, 19], FENS can train the aggregator model on the server. This makes FENS applicable in model market scenarios of OFL [7, 41] where clients primarily offer pre-trained models.

To conclude, we introduce FENS, a hybrid approach combining OFL and FL. FENS emphasizes local training and one-shot model sharing, similar to OFL, which limits communication costs. It then performs lightweight aggregator training in an iterative FL-like fashion. Our experiments on diverse tasks demonstrated that FENS is highly effective in settings with high data heterogeneity, nearly achieving FL accuracy while maintaining the communication efficiency of OFL. Additionally, FENS supports model heterogeneity, rapid unlearning, and is applicable to model markets.

## Acknowledgments

Nirupam is partly supported by Swiss National Science Foundation (SNSF) project 200021_200477, "Controlling The Spread of Epidemics: A Computing Perspective". The authors are thankful to Milos Vujasinovic and Sayan Biswas for their helpful discussions, and to the anonymous reviewers of NeurIPS 2024 for their valuable time and constructive inputs that shaped the final version of this work.

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

**Organization of the Appendix**

# A Datasets

As mentioned in Section 3.1, we focus on classification tasks and experiment with 3 datasets in FLamby benchmark including Fed-Camelyon16, Fed-Heart-Disease and Fed-ISIC2019. Table 5 overviews the selected tasks in this work.

Table 5: Overview of selected datasets and tasks in FLamby. We defer additional details to [32].

| DATASET | INPUT ($x$) | PREDICTION ($y$) | TASK TYPE | # CLIENTS | # EXAMPLES PER CLIENT | MODEL | METRIC |
|---------|-------------|------------------|-----------|-----------|------------------------|-------|--------|
| Fed-Camelyon16 | Slides | Tumor on Slide | Binary Classification | 2 | 239, 150 | DeepMIL [17] | AUC |
| Fed-Heart-Disease | Patient Info. | Heart Disease | Binary Classification | 4 | 303, 261, 46, 130 | Logistic Regression | Accuracy |
| Fed-ISIC2019 | Dermoscopy | Melanoma Class | Multi-class Classification | 6 | 12413, 3954, 3363, 225, 819, 439 | EfficientNet [40] + Linear layer | Balanced Accuracy |

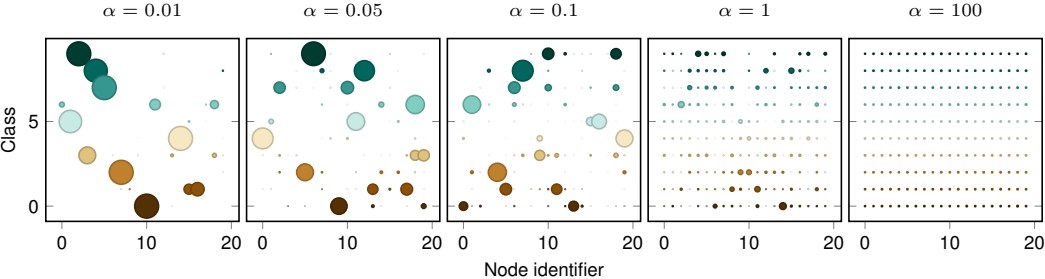

Figure 9: Visualizing the effect of changing $\alpha$ on the CIFAR-10 dataset. Dot size corresponds to the number of samples of a given class in a given node.

# B Additional Experimental Details

## B.1 One-shot FL baselines

The client local training for all OFL baselines (except FEDCVAE-ENS) as well as FENS is conducted alike, using the parameters reported in [10] for the CIFAR-10 and CIFAR-100 datasets. For the SVHN and AG-News datasets, local training is conducted using the SGD optimizer with a learning rate of 0.01 for 50 and 20 local epochs respectively and decayed using Cosine Annealing. All vision datasets use a batch size of 16 while AG-News uses a batch-size of 8 for local training. For FEDCVAE-ENS, we use the same CVAE architecture and local training parameters as reported by the authors [14]. However, for the distillation at the server, we use ResNet-8 as the classifier model at the server to maintain fairness with other baselines. For the distillation phase of FEDKD,

we use the setup described by the authors [10] without inference quantization. For the one-shot version of FED-ET, we set the diversity regularization parameter to the best value of $\lambda = 0.05$. For both FEDKD and FED-ET, we use CIFAR-10, CIFAR-100, and TinyImageNet as the auxiliary datasets for distillation for SVHN, CIFAR-10, and CIFAR-100 datasets respectively. We consider a $60 - 40\%$ split for the AG-News dataset where local training is conducted on the $60\%$ split while the remaining $40\%$ is treated as the auxiliary dataset for distillation at the server in FEDKD. Distillation in Co-Boosting using synthetically generated data also uses the best-reported hyperparameters [7]. For one-round FEDAVG, we additionally tune the number of local epochs performed before aggregation by considering $\{1, 2, 5, 10, 15, 20\}$ epochs.

### B.2 Iterative FL baselines

We experiment with 6 different baselines including FEDAVG, FEDPROX, FEDNOVA, FEDADAM, FEDYOGI, and SCAFFOLD. We perform extensive hyperparameter tuning on the CIFAR-10 dataset for all levels of heterogeneity as detailed below. The server assumes full client participation, *i.e.*, all clients participate in each round. For our vision benchmarks, each client performs 2 local epochs per round using a batch size of 16. For our language task, clients train for 50 local steps in each round using a batch-size of 8. We run FL training until convergence and report the best accuracy achieved. We found all algorithms to converge in less than 100 communication rounds on the vision tasks and in 150 rounds on the language task.

**Hyperparameter tuning.** Below we describe our tuning procedure derived from several previous works [23, 32, 33]. For the FEDAVG algorithm, we tune the client learning rate ($\eta_l$) over the values $\{0.1, 0.01, 0.001, 0.0001\}$ separately for every $\alpha \in \{0.01, 0.05, 0.1\}$. For the FEDPROX algorithm, our grid space was $\{0.1, 0.01\}$ and $\{1, 0.1, 0.01\}$ for the client learning rate ($\eta_l$) and the proximal parameter ($\mu$) respectively. This was again separately tuned for every value of $\alpha \in \{0.01, 0.05, 0.1\}$. For the FEDYOGI and the FEDADAM algorithm, we consider the grid space of $\{0.1, 0.01, 0.001, 0.0001\}$ and $\{10, 1, 0.1, 0.01, 0.001\}$ for the client learning rate ($\eta_l$) and the server learning rate ($\eta_s$) respectively. This explodes the search significantly when tuning for every value of $\alpha$. From our tuning results for the FEDAVG and the FEDPROX algorithm, we noticed that the optimal parameter values were the same within the following two subgroups of $\alpha - \{0.01, 0.05\}$ and $\{0.1\}$ (Table 6). Hence, to keep the tuning tractable, we tune only for one $\alpha$ in each subgroup and reuse the values for other alphas within the same subgroup. For the FEDNOVA algorithm, we use the version with both client and global momentum which was reported to perform the best [43]. We consider the search space $\{0.005, 0.01, 0.02, 0.05, 0.08\}$ for the client learning rate ($\eta_l$) as done by the authors [43] and tune separately for every value of $\alpha$. Finally, for the SCAFFOLD algorithm, we consider the search spaces $\{0.1, 0.01, 0.001, 0.0001\}$ and $\{1.0, 0.1, 0.01\}$ for $\eta_l$ and $\eta_s$ respectively and also tune separately for every $\alpha$. We first conduct the tuning procedure on the CIFAR-10 dataset and report the obtained hyperparameters in Table 6.

Table 6: Best hyperparameters obtained for the different algorithms on the CIFAR-10 dataset.

| | FEDAVG | FEDPROX | | FEDYOGI | | FEDADAM | | FEDNOVA | SCAFFOLD | |
|---|---|---|---|---|---|---|---|---|---|---|
| $\alpha$ | $\eta_l$ | $\eta_l$ | $\mu$ | $\eta_l$ | $\eta_s$ | $\eta_l$ | $\eta_s$ | $\eta_l$ | $\eta_l$ | $\eta_s$ |
| 0.01 | 0.01 | 0.01 | 0.01 | 0.01 | 0.01 | - | - | 0.0001 | 0.0001 | 1.0 |
| 0.05 | 0.01 | 0.01 | 0.01 | - | - | 0.01 | 0.01 | 0.02 | 0.01 | 0.1 |
| 0.1 | 0.1 | 0.1 | 0.01 | 0.01 | 0.01 | 0.01 | 0.01 | 0.005 | 0.01 | 1.0 |

We run all iterative FL baselines using the above parameters and present the results in Table 10. Based on our results in the Table 10 for CIFAR-10, we observe that FEDADAM and FEDYOGI consistently perform the best. Hence, to keep the experiments tractable, we tune and present just FEDADAM as a representative of the iterative FL family for our evaluations on the SVHN, CIFAR-100 datasets in Section 3.3 and AG-News in Section 3.5. For FEDADAM on the AG-News dataset, we note that the training is conducted using only the $60\%$ split (see Appendix B.1) to achieve a fair comparison with FEDKD. For our experiments involving the extended SVHN dataset in Section 3.3.1, we again tune the client learning rate ($\eta_l$) for the FEDAVG algorithm over the search space $\{0.1, 0.01, 0.001, 0.0001\}$ separately for every $\alpha \in \{0.01, 0.05, 0.1\}$.

**FEDAVG with gradient compression.** To implement FEDAVG with gradient compression, we followed the sparsification and quantization schemes of STC [29]. We use the quantization level of 16-bit and sparsity of $50\%$. This results in a communication cost reduction of $4\times$ against standard FEDAVG in every round. For each dataset and heterogeneity level, we tune the learning rate over the search space $\{0.1, 0.05, 0.01, 0.001\}$. We keep the remaining setup the same as FEDAVG.

**Multi-round FEDKD.** Since FENS incurs a communication cost four times that of OFL, we also evaluate FEDKD with multi-round support. We explore two approaches: *i)* pre-training for 3 rounds using FEDAVG, then applying FEDKD, and *ii)* using FEDKD followed by 3 fine-tuning rounds with FEDAVG. In the first case, each FEDKD client begins training from the global model produced by FEDAVG, while in the second, FEDAVG starts from the FEDKD model. We observe that pre-training with FEDAVG offers little improvement, likely due to the forgetting effect from multiple local training epochs, whereas fine-tuning with FEDAVG boosts FEDKD performance. For our experiments in Section 3.3, we thus present the multi-round version of FEDKD with fine-tuning. We remark that this multi-round support is still insufficient to match the performance of FENS, which achieves significantly higher accuracy as shown in Table 7 while incurring similar communication costs.

Table 7: FEDKD under multi-round support on the CIFAR-10 dataset.

| Dataset | $\alpha$ | FEDKD | 3 rounds FEDAVG + FEDKD | FEDKD + 3 rounds FEDAVG | FENS |
|---------|------|-----------------|-----------------|-----------------|-----------------|
| | 0.01 | $18.59_{\pm 2.92}$ | $19.31_{\pm 3.16}$ | $21.81_{\pm 4.27}$ | $\mathbf{44.20}_{\pm \mathbf{3.29}}$ |
| CF-10 | 0.05 | $38.84_{\pm 6.03}$ | $37.92_{\pm 7.04}$ | $43.26_{\pm 7.39}$ | $\mathbf{68.22}_{\pm \mathbf{4.19}}$ |
| | 0.1 | $64.14_{\pm 5.17}$ | $63.25_{\pm 6.22}$ | $62.61_{\pm 6.17}$ | $\mathbf{75.61}_{\pm \mathbf{1.85}}$ |

## B.3 FENS

Clients in FENS perform local training similar to OFL baselines as described in Section 3.1 and appendix B.1. Let $z_j \in \mathbb{R}^C$ denote the logits obtained from each client model $\pi_j$ for all $j \in [M]$ on a given input where $C$ is the number of classes. We use one of the two aggregator models as follows. The first one is a multilayer perceptron using ReLu activations and a final classifier head as follows: $f = W_2^T \sigma(W_1^T z)$ where $W_1 \in \mathbb{R}^{MC \times k}$, $W_2 \in \mathbb{R}^{k \times C}$, $z = \text{concat}(z_1, \ldots, z_M) \in \mathbb{R}^{MC}$ is the concatenated logit vector and $\sigma(x) = \max\{x, 0\}$ is the ReLU function. The parameter $k$ determines the number of units in the hidden layer of this perceptron model. The second one is $f = \sum_{i=1}^{M} \boldsymbol{\lambda}_i \odot z_i$ where $\boldsymbol{\lambda}_1, \ldots, \boldsymbol{\lambda}_M \in \mathbb{R}^C$ are weight vectors and $\odot$ denotes coordinate-wise product. This model learns per-class per-client weights as the model parameters. For all datasets, the aggregator model is trained using the FEDADAM algorithm where the learning rate is separately tuned for each dataset. Table 8 presents the tuned learning rate and tuned training parameters per dataset.

Table 8: Aggregator training in FENS. We use FEDADAM as the FL algorithm with the following client ($\eta_l$) and server ($\eta_s$) learning rates. The parameter $k$ corresponds to the weight matrices $W_1$ and $W_2$.

| Dataset | Aggregator Model | $k$ | $\eta_l$ | $\eta_s$ | Batch Size | Local Steps | Global Rounds |
|---------|------------------|-----|----------|----------|------------|-------------|---------------|
| CIFAR-10 | $f = W_2^T \sigma(W_1^T z)$ | 40 | 1.0 | 0.001 | 128 | 1 | 500 |
| CIFAR-100 | $f = \sum_{i=1}^{M} \boldsymbol{\lambda}_i \odot z_i$ | – | 1.0 | 0.003 | 128 | 1 | 1000 |
| SVHN | $f = W_2^T \sigma(W_1^T z)$ | 40 | 0.1 | 0.01 | 128 | 1 | 500 |
| AG-News | $f = W_2^T \sigma(W_1^T z)$ | 40 | 1.0 | 0.001 | 128 | 1 | 500 |
| Fed-Heart-Disease | $f = \sum_{i=1}^{M} \boldsymbol{\lambda}_i \odot z_i$ | – | 0.1 | 0.1 | 2 | 5 | 50 |
| Fed-Camelyon16 | $f = \sum_{i=1}^{M} \boldsymbol{\lambda}_i \odot z_i$ | – | 1.0 | 0.0005 | 64 | 1 | 2000 |
| Fed-ISIC2019 | $f = W_2^T \sigma(W_1^T z)$ | 24 | 1.0 | 0.001 | 16 | 1 | 2500 |

**Model quantization in FENS.** Clients in FENS incur a critical cost of downloading the ensemble model from the server to initiate the aggregator training. To reduce the communication burden on the clients, the server employs post-training model quantization of all received client local models from `FP32` to `INT8`, reducing the download costs by $4\times$. Alternatively, the quantization can also be executed on the client side. The quantization results in a drop of $\approx 1 - 2\%$ test accuracy for every client model compared to the corresponding non-quantized model. However, the subsequent

aggregator training phase in FENS provides resilience to this drop in the accuracy of client models in the ensemble. In fact, we observe that the final accuracy achieved after aggregator training is slightly higher when using the quantized models as compared to unquantized models due to the regularising effect of quantization on generated logits. The model quantization also provides reduced memory usage on client devices during aggregator training. We use the standard PyTorch quantization library to implement quantization in FENS.

## B.4 Aggregation rules

**Averaging.** Averaging corresponds to the following static aggregation rule: $f = \sum_{i=1}^{M} \boldsymbol{\lambda}_i \odot \boldsymbol{z}_i$ where $\boldsymbol{\lambda}_i = [\frac{1}{M}, \ldots, \frac{1}{M}]$ and $\odot$ denotes coordinate-wise product.

**Weighted Averaging.** In weighted averaging, the $\boldsymbol{\lambda}_1, \ldots, \boldsymbol{\lambda}_M \in \mathbb{R}^C$ are typically assigned based on local training dataset statistics. In FEDKD [10], $\boldsymbol{\lambda}_i = [\frac{n_i^1}{\sum_{i \in [M]} n_i^1}, \frac{n_i^2}{\sum_{i \in [M]} n_i^2}, \ldots, \frac{n_i^C}{\sum_{i \in [M]} n_i^C}]$ where $n_i^j$ corresponds to the number of samples of class $j$ with client $i$.

**Linear.** This aggregation corresponds to having a single learnable scalar weight for each client $f = \sum_{i=1}^{M} w_i \boldsymbol{z}_i$. The learnable parameters in this case consist of the vector $[w_1, w_2, \ldots, w_M]^T$.

**Neural network (NN).** Let $\boldsymbol{z}_j \in \mathbb{R}^C$ denote the logits obtained from each client model $\pi_j$ for all $j \in [M]$ on a given input where $C$ is the number of classes. The NN aggregation corresponds to any neural network-based model $f : \mathcal{Z}^M \to \mathcal{Z}$ that operates on the logits produced by the client models. Denoting $\boldsymbol{z} = \text{concat}(\boldsymbol{z}_1, \ldots, \boldsymbol{z}_M) \in \mathbb{R}^{MC}$ as the concatenated vector of logits, $f$ corresponds to the following 2 layer neural network $f = \boldsymbol{W}_2^T \sigma(\boldsymbol{W}_1^T \boldsymbol{z})$ where $\boldsymbol{W}_1 \in \mathbb{R}^{MC \times k}$, $\boldsymbol{W}_2 \in \mathbb{R}^{k \times C}$ and $\sigma(x) = \max\{x, 0\}$ is the ReLU function. Here $k$ determines the number of units in the hidden layer and controls the expressivity of the network. This aggregation is much more powerful than the previously mentioned aggregations, owing to its ability to discern complex patterns across all $M \times C$ logits. The learnable parameters comprise the weight matrices $\{\boldsymbol{W}_1, \boldsymbol{W}_2\}$.

**Polychotomous Voting.** Polychotomous voting [2], was originally developed in social choice theory to reach a collective decision when offered $C$ alternatives (classification labels in our case) in a committee of $M$ experts (clients in our case). This method requires as input: *(i)* classwise "competency" scores of each client: $P_i^c(r)$ indicating the probability of $i^{th}$-client to vote for label $c$ when the ground truth is $r$; *(ii)* $p_{prior}(r)$ : prior probability distribution over correct alternatives $r$; and *(iii)* the "benefit" vector of the committee: $B(c|r)$ indicating the committee's benefit in choosing label $c$ when the correct class is $r$. Given this information, Ben-Yashar and Paroush [2] derive a criterion for the optimal decision that maximizes expected utility. This criterion is not computed using a closed-form expression, and we generically express it as

$$f(\pi_1, \ldots, \pi_M; \boldsymbol{P}_1, \ldots, \boldsymbol{P}_M; \boldsymbol{p}_{prior}; B) \tag{3}$$

where $f$ corresponds to a procedure that evaluates and compares benefits for each choice $c \in [C]$ given the competency matrices $\{\boldsymbol{P}_i\}_{i=1}^M$, the priors $\boldsymbol{p}_{prior}$ and the benefit function $B$. Since the competency matrices for each client model are not directly available in our distributed setting, we learn them by federation in the network. More specifically, each client computes the competency matrix for every client model in the ensemble on its local data and transfers them to the server. The server then aggregates the received competency matrix to produce the final competency matrix per client to be used in decision-making. We further use a simple benefit function for our experiments that assigns $B(c/r) = 1$ when $c = r$ (correct choice) and $B(c/r) = 0$ when $c \neq r$ (incorrect choice) and set the prior to be uniform over the set of labels.

**Mixture-of-Experts (MoE).** The MoE aggregation [36] considers both the input and the logits in the following form: $f = \sum_{i \in [M]} G(x)_i . \pi_i(x)$. Thus, MoE aggregation $f : (\mathcal{X}, \mathcal{Z}^M) \to \mathcal{Z}$ is more expressive compared to other aggregations which only consider logits $f : \mathcal{Z}^M \to \mathcal{Z}$. Here, $G : \mathcal{X} \to [0, 1]^M$ is called a gating network that generates scalar weights for every expert $\pi_1, \ldots, \pi_M$ based on the input $x \in \mathcal{X}$. In FENS with MoE aggregation, only the gating network is trained via the federation in the network while $\{\pi_i\}_{i=1}^M$ correspond to the locally trained client models. We use a simple CNN with two convolutional blocks comprising ReLU activation and max pooling layers followed by 2 FC layers with ReLU activation, which in turn is followed by the final classification head. Despite its expressivity, learning a good gating network incurs significant communication costs and remains difficult under heterogeneous data in federated settings.

---
**Algorithm 1:** FENS from server perspective
---
**Require :** $M$ clients, boolean *quantize*
**1 Procedure** FENS_SERVER*()*:
**2**     Initialize model $\pi$ with parameters $\theta$ for local training
**3**     Send $\theta$ to all $M$ clients
**4**     Receive parameters of locally trained models $\{\theta^{(i)},\ i \in [M]\}$
**5**     **if** *quantize* **then**
**6**        $\theta^{(i)} \leftarrow$ `quantization_alg`$(\theta^{(i)}, \text{FP32}, \text{INT8})\ \forall i \in [M]$
**7**     Send $\{\theta^{(i)},\ i \in [M]\}$ to all $M$ clients[1]
**8**     Initialize aggregator model $f_\lambda$ with parameters $\lambda_0$
**9**     **for** $t = 0, 1, \ldots$ *until convergence* **do**
**10**        Select $S_t \subseteq [M]$ and send them the aggregator parameters $\lambda_t$
**11**        Receive updated parameters $\{\lambda_t^{(i)},\ i \in S_t\}$
**12**        Update global aggregator model $\lambda_{t+1} := \frac{1}{|S_t|} \sum_{i \in S_t} \lambda_t^{(i)}$[2]
---

---
**Algorithm 2:** FENS from the clients perspective
---
**Require :** Local dataset $\mathcal{D}_i$, loss function $\ell$, local steps $K$ and client learning rate $\eta_l$
**1 Procedure** FENS_CLIENT*()*:
**2**     Split $\mathcal{D}_i$ randomly into 90% $\mathcal{D}_{i1}$ and 10% $\mathcal{D}_{i2}$
**3**     Receive parameters $\theta$ from the server
**4**     Obtain $\theta^{(i)}$ through local training of $\theta$ on $D_{i1}$
**5**     Send converged model parameters $\theta^{(i)}$ to server (one-shot)
**6**     Receive $\{\theta^{(i)},\ i \in [M]\}$ from the server
**7**     **while** *Receive aggregator model parameters $\lambda_t$ from the server* **do**
**8**        Initialize $\lambda_t^{(i)} \leftarrow \lambda_t$
**9**        **for** $k = 0, 1, \ldots, K$ **do**
**10**           Sample mini-batch $b \in \mathcal{D}_{i2}$
**11**           $\ell_b \leftarrow \frac{1}{|b|} \sum_{(x,y) \in b} \ell\left( f_{\lambda_t^{(i)}}(\pi_1(x), \ldots, \pi_M(x)), y \right)$
**12**           $\lambda_t^{(i)} \leftarrow \lambda_t^{(i)} - \eta_l \nabla \ell_b$
**13**        Send $\lambda_t^{(i)}$ back to the server
---

## C   FENS Algorithm

Algorithm 1 outlines the role of the server in FENS. The process begins with the server initializing the parameter $\theta$ corresponding to the parametric model $\pi = h_\theta$, which it sends to all clients for local training (lines 2-3). Once clients complete their local training, they return their updated models to the server (line 4). If quantized is enabled, the server quantizes all the local models from FP32 to INT8 using a quantization algorithm (line 6). The server then redistributes all models back to the clients (line 7), enabling each to contribute to the aggregation process that follows. In the final stage, the server iteratively trains an aggregator model in FL fashion, which is designed to combine client models into a single, improved global model (lines 9-12). During each round, the server selects a subset of clients to update the aggregator model (line 10), refining it further with each iteration until convergence.

Algorithm 2 explains the client-side process in FENS. Each client starts by splitting its local dataset into two parts: one for one-shot local training and a smaller part for the iterative aggregator training (line 2). Using the received model, clients first train on $\mathcal{D}_{i1}$ (line 4) and send their converged local model back to the server (line 5). In subsequent rounds, clients receive from the server the aggregator model (line 7) which they refine it locally using $\mathcal{D}_{i2}$ (lines 8-12). Finally, clients send the updated aggregator parameters back to the server (line 13), contributing to the global aggregation process.

---
[1]The server can potentially send all parameters except the client's own to reduce costs.
[2]The server can use any FL algorithm for aggregator training, FEDAVG shown for simplicity.

## D  Numerical Results

In this section, we include the numerical values in Tables 9 to 17 corresponding to the plots presented in Sections 3.3, 3.4 and 3.6 for a complete reference.

Table 9: FENS aggregation methods on CIFAR-10. Results of Figure 8.

| Algorithm | $\alpha = 0.01$ | $\alpha = 0.05$ | $\alpha = 0.1$ |
|---|---|---|---|
| Averaging | $15.66_{\pm 6.11}$ | $39.56_{\pm 6.33}$ | $48.40_{\pm 9.01}$ |
| Weighted Averaging | $17.62_{\pm 3.38}$ | $37.72_{\pm 6.35}$ | $62.55_{\pm 6.03}$ |
| Polychotomous Voting | $26.32_{\pm 2.54}$ | $52.28_{\pm 3.42}$ | $63.23_{\pm 1.12}$ |
| Linear Aggregator | $27.03_{\pm 6.50}$ | $56.94_{\pm 6.50}$ | $67.64_{\pm 4.52}$ |
| NN Aggregator | $44.20_{\pm 3.29}$ | $\mathbf{68.22}_{\pm 4.19}$ | $\mathbf{75.61}_{\pm 1.85}$ |
| MoE (Gating) | $\mathbf{49.86}_{\pm 1.61}$ | $62.11_{\pm 3.62}$ | $69.50_{\pm 2.61}$ |

Table 10: FENS vs SOTA FL algorithms on the CIFAR-10 dataset.

| Algorithm | $\alpha = 0.01$ | $\alpha = 0.05$ | $\alpha = 0.1$ |
|---|---|---|---|
| FEDADAM | $39.324_{\pm 7.855}$ | $\mathbf{68.748}_{\pm 2.762}$ | $78.736_{\pm 3.552}$ |
| FEDAVG | $37.600_{\pm 6.428}$ | $60.344_{\pm 1.705}$ | $77.062_{\pm 3.678}$ |
| FEDNOVA | $32.316_{\pm 3.844}$ | $60.280_{\pm 2.300}$ | $78.732_{\pm 3.252}$ |
| FEDPROX | $37.344_{\pm 6.001}$ | $59.772_{\pm 1.413}$ | $75.250_{\pm 4.705}$ |
| FEDYOGI | $39.788_{\pm 7.726}$ | $67.168_{\pm 2.793}$ | $\mathbf{78.980}_{\pm 3.047}$ |
| SCAFFOLD | $21.824_{\pm 3.168}$ | $27.308_{\pm 8.303}$ | $65.892_{\pm 23.98}$ |
| FENS | $\mathbf{44.200}_{\pm 3.297}$ | $68.220_{\pm 4.197}$ | $75.613_{\pm 1.858}$ |

Table 11: FENS vs. iterative FL. Results from Figure 4.

| Dataset | $\alpha$ | FEDKD 1R | FEDAVG STC | FEDKD 4R | FENS | FEDADAM (Max acc.) |
|---|---|---|---|---|---|---|
| CF-100 | 0.01 | $28.98_{\pm 4.55}$ | $24.83_{\pm 1.27}$ | $32.22_{\pm 2.39}$ | $44.46_{\pm 0.31}$ | $50.25_{\pm 0.58}$ |
| | 0.05 | $39.01_{\pm 1.11}$ | $29.67_{\pm 0.46}$ | $42.12_{\pm 0.23}$ | $49.70_{\pm 0.86}$ | $56.62_{\pm 0.28}$ |
| | 0.1 | $42.38_{\pm 0.78}$ | $31.87_{\pm 0.58}$ | $42.66_{\pm 0.89}$ | $51.11_{\pm 0.37}$ | $59.96_{\pm 0.80}$ |
| CF-10 | 0.01 | $18.59_{\pm 2.92}$ | $22.70_{\pm 7.08}$ | $21.81_{\pm 4.27}$ | $44.20_{\pm 3.29}$ | $39.32_{\pm 7.85}$ |
| | 0.05 | $38.84_{\pm 6.03}$ | $50.56_{\pm 3.60}$ | $43.26_{\pm 7.39}$ | $68.22_{\pm 4.19}$ | $68.74_{\pm 2.76}$ |
| | 0.1 | $64.14_{\pm 5.17}$ | $61.93_{\pm 3.01}$ | $62.61_{\pm 6.17}$ | $75.61_{\pm 1.85}$ | $78.73_{\pm 3.55}$ |
| SVHN | 0.01 | $23.62_{\pm 10.1}$ | $32.64_{\pm 12.9}$ | $25.82_{\pm 6.96}$ | $57.35_{\pm 12.6}$ | $66.85_{\pm 9.26}$ |
| | 0.05 | $37.41_{\pm 9.62}$ | $75.48_{\pm 4.39}$ | $49.21_{\pm 8.17}$ | $76.76_{\pm 2.98}$ | $83.55_{\pm 1.52}$ |
| | 0.1 | $61.38_{\pm 3.90}$ | $85.32_{\pm 2.66}$ | $74.33_{\pm 3.34}$ | $83.64_{\pm 0.75}$ | $88.04_{\pm 0.06}$ |

Table 12: Figure 6 results. FENS vs. iterative FL – Fed-Camelyon16 (row 1).

| Algorithm | AUC |
|---|---|
| FEDADAM | $0.500_{\pm 0.000}$ |
| FEDAVG | $0.569_{\pm 0.063}$ |
| FEDPROX | $0.584_{\pm 0.069}$ |
| FEDYOGI | $0.513_{\pm 0.026}$ |
| SCAFFOLD | $0.485_{\pm 0.100}$ |
| FENS | $\mathbf{0.715}_{\pm 0.024}$ |

Table 13: Figure 6 results. FENS vs. one-shot FL – Fed-Camelyon16 (row 2).

| Algorithm | AUC |
|---|---|
| Client 0 | $0.696_{\pm 0.018}$ |
| Client 1 | $0.683_{\pm 0.067}$ |
| FEDAVG-OS | $0.673_{\pm 0.007}$ |
| FEDPROX-OS | $0.618_{\pm 0.043}$ |
| FENS | $\mathbf{0.715}_{\pm 0.024}$ |

Table 14: Figure 6 results. FENS vs. iterative FL – Fed-Heart-Disease (row 1).

| Algorithm | Accuracy |
|---|---|
| FEDADAM | $0.781_{\pm 0.002}$ |
| FEDAVG | $\mathbf{0.794}_{\pm 0.004}$ |
| FEDPROX | $0.792_{\pm 0.009}$ |
| FEDYOGI | $0.785_{\pm 0.002}$ |
| SCAFFOLD | $0.792_{\pm 0.006}$ |
| FENS | $0.781_{\pm 0.027}$ |

Table 15: Figure 6 results. FENS vs. one-shot FL – Fed-Heart-Disease (row 2).

| Algorithm | Accuracy |
|---|---|
| Client 0 | $\mathbf{0.796}_{\pm 0.007}$ |
| Client 1 | $0.750_{\pm 0.035}$ |
| Client 2 | $0.553_{\pm 0.059}$ |
| Client 3 | $0.599_{\pm 0.047}$ |
| FEDAVG-OS | $0.698_{\pm 0.048}$ |
| FEDPROX-OS | $0.732_{\pm 0.014}$ |
| FENS | $0.781_{\pm 0.027}$ |

Table 16: Figure 6 results. FENS vs. iterative FL – Fed-ISIC2019 (row 1).

| Algorithm | Balanced Accuracy |
|---|---|
| FEDADAM | $0.710_{\pm 0.011}$ |
| FEDAVG | $0.731_{\pm 0.007}$ |
| FEDPROX | $0.750_{\pm 0.006}$ |
| FEDYOGI | $0.731_{\pm 0.020}$ |
| SCAFFOLD | $\mathbf{0.746}_{\pm 0.008}$ |
| FENS | $0.579_{\pm 0.022}$ |

Table 17: Figure 6 results. FENS vs. one-shot FL – Fed-ISIC2019 (row 2).

| Algorithm | Balanced Accuracy |
|---|---|
| Client 0 | $0.573_{\pm 0.007}$ |
| Client 1 | $0.315_{\pm 0.012}$ |
| Client 2 | $0.433_{\pm 0.005}$ |
| Client 3 | $0.325_{\pm 0.014}$ |
| Client 4 | $0.181_{\pm 0.008}$ |
| Client 5 | $0.217_{\pm 0.012}$ |
| FEDAVG-OS | $0.424_{\pm 0.007}$ |
| FEDPROX-OS | $0.421_{\pm 0.017}$ |
| FENS | $\mathbf{0.579}_{\pm 0.022}$ |

# E   Compute Resources

We use a cluster comprising a mix of 2x Intel Xeon Gold 6240 @ 2.5 GHz of 36 hyper-threaded cores and 2x AMD EPYC 7302 @ 3 GHz of 64 hyper-threaded cores, equipped with 4x NVIDIA Tesla V100 32G and 8x NVIDIA Tesla A100 40G GPU respectively. Training of local models can take up to 2.5 hours in wall-clock time depending on the dataset, while FENS aggregator model training executes in under 30 minutes in wall-clock time. The time required for executing the baselines varies significantly depending on the baseline, with up to 24 hours in wall clock time for Co-Boosting. Across all experiments that are presented in this article, including different seeds and hyperparameter tuning, the total virtual CPU and GPU time is approximately 3500 and 8000 hours respectively.

# F   Broader Impact

Federated Learning (FL) has significantly advanced privacy-preserving machine learning, particularly in sensitive domains like healthcare and finance, by facilitating collaborative model training without sharing raw data. The development of FENS, which combines FL's accuracy with the communication efficiency of One-shot FL (OFL), carries numerous positive implications. By simultaneously reducing communication costs and maintaining high accuracy, FENS enhances the accessibility and practicality of FL, thereby promoting wider adoption, especially in resource-constrained environments. This has the potential to catalyze advancements in privacy-sensitive sectors like healthcare and finance, where FL is extensively utilized.

