# OpenReview forum: "Revisiting Ensembling in One-Shot Federated Learning"
_NeurIPS.cc/2024/Conference — NeurIPS 2024 poster_

### Official Review · Reviewer_y6N3 · 2024-06-19

**Soundness:** 3
**Presentation:** 3
**Contribution:** 2
**Rating:** 5
**Confidence:** 4

**Summary:**

This work proposes a one-shot federated learning method where, instead of simply aggregating the models at the end of training, a non-linear ensemble of the (frozen) models is trained with an iterative federated learning method. In other words, after training the models locally, the server trains a shallow neural network that takes the logits of all the models as input and outputs the aggregated output. This approach is feasible in the cross-silo setting. The experiments show that it can be significantly better than SOTA one-shot methods and it can sometimes approach the performance of full federated averaging (i.e., with multiple rounds of aggregation).

**Strengths:**

- The approach is simple yet highly efficient in the heterogenous cross-silo settings, as shown in Figure 2.
- The paper is well-written and the experiments cover most of the interesting cases.
- The authors share the code for reproducing the results, which further improves the quality of the submission. The code itself is well written and reproducibility is made easy given the clear instructions.
- The results are demonstrated on various datasets, including real-world health data.
- The discussion regarding client unlearning is interesting and does show a desirable strength in the proposed method.
- The authors clearly mention the limitations and propose future directions to explore and mitigate some of the issues.

**Weaknesses:**

The contribution of the paper is a bit limited in terms of technical novelty. I will not base my rating on novelty alone, of course. However, the technical details of the method are a bit straightforward and not new (e.g., the authors mention Stacked Generalization by Wolpert (1992)). Nonetheless, I will try to be fair and rate this paper's contribution in terms of its effectiveness and (reproducible) experimental results, which can be as important as technical novelty.

- Collecting the logits of all clients for every forward pass of the ensemble is not elegant, especially given the memory and computational costs. The authors do acknowledge this limitation and offer some directions to mitigate this, but they still do not offer a pratical solution that scales well. The authors mentioned quantization but did not share any preliminary findings or insights. I think running experiments where some clients are sampled every round of training the ensemble is straightfoward to implement, so it would be interesting to see that. Also, how would you concatenate the logits?
- It might be somewhat fair to compare with one or two personalized FL algorithms, since the proposed method already assumes a large memory budget per client, so such clients in the cross-silo setting are also quite likely to be able to maintain some state (or at least, have the server maintain it for them).
- Another intuitive thing to compare to is split learning methods, where some of the layers are trained with iterative FL methods, and the others are kept local. These methods might be able to reach similar accuracies with a more feasible memory and computational budget.

**Questions:**

- Does the communication cost comparison in Figure 2 assume quantization? If so, does the communication cost for the iterative FL methods also use quantization?
- The performances reported in Appendix D are impressive, but Table 15 (Fed-ISIC2019) shows a limitation/non-robustness of one-shot methods on a real-world dataset. Why do you think the gap here is large?

**Limitations:**

The authors adequately addressed the limitations. There are no negative societal impacts associated with the proposed method.

---

> ### Author Rebuttal · Authors · 2024-08-06
>
> We thank the reviewer for their positive feedback and constructive comments. We address the reviewer's questions below.
>
> --------------
> > Q. Does the communication cost comparison in Figure 2 assume quantization? If so, does the communication cost for the iterative FL methods also use quantization?
>
> $\rightarrow$ Yes, FENS is presented with quantization in Figure 2.
> While the iterative FL baseline (FedAdam) in Figure 2 does not use quantization, we have included the relevant baseline with quantization, called FedAvg STC in Figure 4.
> The quantization in FedAvg STC results in an accuracy drop compared to FedAdam.
>
> --------------
> > Q. The performances reported in Appendix D are impressive, but Table 15 (Fed-ISIC2019) shows a limitation/non-robustness of one-shot methods on a real-world dataset. Why do you think the gap here is large?
>
> $\rightarrow$ This gap can be explained through the quality of local models in case of the FedISIC2019 dataset together with our experimental results from Section 3.3.1.
> Our experiments in Section 3.3.1 show that local data size plays a key role in determining when FENS can match the performance of iterative FL.
> The FedISIC2019 dataset, in particular, exhibits high variance in the amount of local data held by the clients, and consequently their local model performance.
> As reported in Table 3 (Appendix A), the biggest client has 12k samples while the smallest client has only 225 samples.
> Thus, we speculate that the FedISIC2019 dataset falls in the low local training fraction regime of the curves in Fig. 5, exhibiting a larger gap compared to iterative FL.
>
> --------------
> > Collecting the logits of all clients for every forward pass of the ensemble is not elegant, especially given the memory and computational costs. The authors do acknowledge this limitation and offer some directions to mitigate this, but they still do not offer a practical solution that scales well.
>
> $\rightarrow$ We currently use FP32 to INT8 quantization to mitigate this issue (Appendix B.3).
> This reduces the costs by $4\times$.
> An alternative and complementary approach is to consider an ensemble such that the total size is comparable to the single model trained in iterative FL.
> Preliminary results indicate that under a comparable memory footprint, FENS still remains competitive with the original FedAdam.
> We will include the complete ablation in our revised version.
>
> --------------
> > I think running experiments where some clients are sampled every round of training the ensemble is straightforward to implement, so it would be interesting to see that. How would you concatenate the logits?
>
> $\rightarrow$ This is indeed a highly relevant suggestion.
> We explored sampling clients each round by setting logits from non-sampled models to zero in the concatenated vector.
> Additionally, we also tried learning mask tokens that fill for non-sampled logits.
> Preliminary results show an accuracy drop under high heterogeneity, likely because each model’s contribution is significant in such scenarios.
> To improve performance, a more sophisticated logit concatenation method may be required.
>
> --------------
> > On suggested comparison to personalized FL baselines and Split Learning.
>
> $\rightarrow$ We thank the reviewer for their insightful suggestions.
> We agree that Split Learning might help make the computational budget more feasible.
> In FENS, we focus on simultaneously achieving OFL-like communication efficiency and FL-like accuracy.
> In this sense, the objectives of Split Learning and personalized FL slightly differ for a direct comparison.
> Personalized FL focuses on individual client accuracy, while FENS targets global data accuracy.
> Similarly, Split Learning reduces computational burden, whereas FENS prioritizes communication efficiency.
> We believe that both these paradigms can be orthogonally explored within FENS and are keen on exploring these aspects in future work.

---

> > ### Comment · Reviewer_y6N3 · 2024-08-12
> >
> > Thanks for your response. I think it would be interesting to see whether such a sophisticated logit concatenation method exists or not. I also agree with Reviewer zQJt that evaluations on data with more complex heterogeneities would be nice to see.

---

> > > ### Author Response · Authors · 2024-08-13
> > >
> > > Thank you for your response and constructive comments in these busy hours. We are grateful for your feedback and look forward to exploring these aspects in our future work.

---

### Official Review · Reviewer_B5VB · 2024-06-26

**Soundness:** 4
**Presentation:** 4
**Contribution:** 2
**Rating:** 6
**Confidence:** 5

**Summary:**

This paper introduces FENS, a One-shot FL (OL) approach with the aim of improving the globl model's accuracy without significantly increasing the communication cost of canonical OFL. Different from existing OFL methods, FENS employs and iteratively trains a prediction aggregator model stacked on top of the local model ensemble.

**Strengths:**

- The method is easy to reproduce without requiring addtional training data on the server.
- Extensive experiments were performed to showcase the claimed performance from multiple aspects.

**Weaknesses:**

- Lack of theoretical analysis to ground the design of the proposed method.
- FENS requires dispatching the whole set of local models to every client in the system, which raises privacy concerns.

**Questions:**

1. How to determine the best number of iterations for the aggregator model?
2. How is the MoE method implemented over the local model ensemble? Why did it incur such a high cost?

**Limitations:**

As mentioned in the Weaknesses section.

---

> ### Author Rebuttal · Authors · 2024-08-06
>
> We thank the reviewer for their positive feedback and  comments. We address the reviewer's questions below.
>
> --------------
> > Q. How to determine the best number of iterations for the aggregator model?
>
> $\rightarrow$ This is determined as in standard FL schemes by stopping when the validation loss has converged or validation accuracy is not improving further.
>
> --------------
> > Q. How is the MoE method implemented over the local model ensemble? Why did it incur such a high cost?
>
> $\rightarrow$ MoE aggregation involves both the input and the logits in the following form: $f(x)=\sum_{i \in [M]}G(x)\_i\pi_i(x)$. Here, $G: \mathcal{X} \rightarrow [0,1]^M$ is the gating network that generates scalar weights for every expert $[\pi_1,\ldots,\pi_M]$  based on the input $x \in \mathcal{X}$.
> In FENS with MoE aggregation, only the gating network is trained via the federation while $[\pi_1,\ldots,\pi_M]$  correspond to the (frozen) locally trained client models.
> For the gating network, we employ a CNN with two convolutional blocks, followed by two fully connected layers and a final classification head.
> Since the gating model has several layers, it incurs a bigger communication overhead of iterative transfers as compared to just an MLP based shallow aggregator (called NN aggregator) in Fig. 7.
> We also tried using smaller models, however these resulted in accuracy deterioration.
> We have included the MoE training details in Appendix B.4.
>
> --------------
> > Lack of theoretical analysis to ground the design of the proposed method.
>
> $\rightarrow$  We acknowledge that deriving a formal theoretical explanation for our empirical findings is an interesting and valuable research direction.
> However, as briefly discussed in Section 2.2, our algorithm is motivated by the study of stacked generalization [1] in the centralized ensemble literature.
> It has been demonstrated that such stacking of models leads to higher-level models correcting for the biases of lower-level models in the stack, thereby improving overall generalization performance.
> While stacked generalization has been primarily studied in centralized setting, through FENS, we demonstrate that this scheme can be efficiently realized under the communication constraints of FL through our novel two-phase training procedure.
>
> --------------
> > Dispatching the whole ensemble to clients raises privacy concerns.
>
> $\rightarrow$
> We agree that privacy is an important concern.
> We would like to note however that FENS still does not require the clients to transfer their local data but only their fully trained models.
> Exploring how to ensure that these model exchanges do not leak information about the data is of independent interest, and could have applications in various FL contexts.
> While we leave an in-depth exploration for future work, we believe that existing privacy schemes could be directly incorporated into FENS.
> For instance, local model training can be done using differentially private SGD [2], thereby ensuring differential privacy of local data despite the sharing of the local models.
> Similarly, the aggregator training can be conducted via a differentially private FL algorithm [3].
> We will include this discussion in our revised version.
>
> --------------
> [1] David H. Wolpert. Stacked generalization. Neural Networks, 5(2):241–259, 1992.
>
> [2] Abadi, Martin, et al. "Deep learning with differential privacy." Proceedings of the 2016 ACM SIGSAC conference on computer and communications security. 2016.
>
> [3] Noble, Maxence, et al. "Differentially Private Federated Learning on Heterogeneous Data". Proceedings of the 25th International Conference on Artificial Intelligence and Statistics (AISTATS). 2022.

---

### Official Review · Reviewer_zQJt · 2024-07-12

**Soundness:** 3
**Presentation:** 3
**Contribution:** 3
**Rating:** 7
**Confidence:** 4

**Summary:**

This paper proposes a 2-phase mechanism for learning models across clients in federated learning settings with minimal communication. The method, FENS, first uses a one-shot communication to learn copies of a base model across clients' local data. It then uses standard iterative FL approaches to learn a smaller "aggregator model", which aggregates some output across the base models.

The authors provide experiments on FENS, comparing it to one-shot approaches and iterative approaches to FL. Generally, their results show that FENS is superior to other methods when judged in accordance with the total amount of communication, and can also achieve higher accuracy (essentially via ensembling) than other methods with reduced total communication.

**Strengths:**

I think this is a well-written paper. The authors are very clear about how the method operates, how it fits into the greater landscape of FL methods, and their experiments are similarly quite clear in execution.

Of course, the core of the work (and content-wise, most of the work) is in the empirical evaluation of FENS. There are a lot of laudable qualities to their empirical analysis. First, I like that the authors compare FENS to one-shot & iterative methods separately. The metrics of interest are different between these settings, and I think it's important to perform different analyses in such cases. I also appreciated the fact that the authors explicitly do not try to claim that FENS is optimal for every settings - the ablation study in Section 3.3.1 is a really good and honest one, that tries to understand the utility of FENS if accuracy is the predominating concern, not communication-effiicency.

I also really appreciated the use of the FLamby benchmark. Synthetically partitioned datasets like CIFAR are only of limited utility in evaluating methods, and the FLamby benchmark is a great choice for datasets that have inherent, realistic forms of heterogeneity. In fact, I think that FLamby would make a stronger benchmark for the other ablation studies (though more on that below).

Last, I really liked Section 3.5. I think the comparisons between FENS and things like mixture-of-experts is a natural thing to consider, and I found this section intriguing enough that I actually wish it were explored more.

**Weaknesses:**

There are some weaknesses to the paper, though I would consider them more as opportunities for improvement. In case it was not obvious from my assessment of the strengths, I think this is a good paper and should be accepted. I break these areas up separately below.

### Limitations of synthetically partitioned datasets & vision datasets

Most of the experiments in this work are on SVHN, CIFAR-10, and CIFAR-100. While these are often used in federated learning research, they are quite limiting. The heterogeneity you are able to study with them is generally limited to label heterogeneity (e.g. due to the Dirichlet mechanism employed). This is only one type of heterogeneity, and generally these tasks are still simpler than realistic, very heterogeneous datasets. FLamby is a great example of realistic heterogeneity, but there are other such datasets, including iNaturalist, GLD-v2, and FLAIR.  I think that these would constitute stronger benchmarks for the work.

Second, I will note that FENS does not specifically require vision tasks. Especially when considering analogies to the mixture-of-experts literature, it is not clear to me why there is no evaluation on a language model task. Language datasets often exhibit interesting heterogeneity and challenge in optimization.

### Trimming Sections 1 & 2 and expanding other discussions

To be honest, I think that Sections 1 & 2 are somewhat repetitive, and span nearly 3.5 pages before any of the actual content of the work comes to bear. The contributions section itself is more than half a page, and Figure 1 does not give any real insight into FENS that isn't already in the text.

By contrast, the actual discussion of the aggregator model architecture is in need of expanding. For example, what architectures would actually be useful for an aggregator model? Why use a 2-layer ReLU network? Could you instead view the client models as a large "backbone model" and view the entire combination of backbone + aggregator as a large model? What happens if you train this large model directly? These are all examples of questions that I think are not addressed, and would be much more interesting than repeating the details of FENS multiple times.

### Considering other system metrics & ablations

The primary metric of interest in this work is total communication. This is a perfectly reasonable metric, but it does obscure other metrics of interest. For example, and as is briefly hinted at in Section 4, FENS incurs a large overhead in terms of server memory as it stores all client models separately. By contrast, the iterative FL methods train a single base model. This means that FENS' memory overhead is more than $M\times$ as much (where $M$ is the number of clients).

What would happen if we equalized server memory usage, so that FENS' base model would be smaller? What happens if we enlarge the models trained by methods like FedAdam? For that matter, what if we measure total client computation, instead of communication costs? Some of these are interesting questions that can actually be answered in part by simply taking a different view of the experiments the authors have already performed. I think that giving these ablations would be hugely interesting, and make the paper much better.

Note that this dovetails with the limitation above - that Sections 1 & 2 are too long. By contrast, Appendix C is interesting, and should probably be added to the main body.

### Data minimization and privacy

One drawback of FENS that is not really discussed (except in L333) is its lack of compatibility with privacy-preserving technologies. The separate one-shot trained base models are stored separately. This means that techniques like differential privacy are not directly compatible. This is a sever weakness of the method. While I don't think it means the paper isn't interesting, I think it should be discussed. Are there potential future directions that could help avoid this issue? Given the fact that FL is predicated on data minimization & privacy, I think some more up front discussion about this would be useful.

**Questions:**

1. How do the results in Figures 2 and 4 change if we instead consider total client computation instead?
1. Why does the MoE approach to aggregation require such a large communication overhead? Can't the size of the dense layers used for routing be tuned?
1. What happens if you apply a method like FedAdam to a model whose total size is comparable to the total FENS model size?

**Limitations:**

The checklist is adequate.

---

> ### Author Rebuttal · Authors · 2024-08-06
>
> We thank the reviewer for their positive assessment of our work and the suggestions for improvements.
> We address the reviewer's questions below.
>
> -----------------
> > Q. Why does the MoE approach to aggregation require such a large communication overhead? Can't the size of the dense
> layers used for routing be tuned?
>
> $\rightarrow$ Yes, we considered tuning the size of the routing network.
> We had experimented with models of various sizes, and observed that smaller models cause a deterioration in accuracy.
> Our current setup employs a CNN with two convolutional blocks, followed by two fully connected layers and a final classification head (Appendix B.4). This configuration achieves a reasonable balance between accuracy and communication overhead.
> Since the routing network needs to process images as input, it must be a reasonably sized CNN.
> Consequently, this results in a larger communication overhead compared to using a shallow MLP-based aggregator (referred to as the NN aggregator in Fig. 7), which processes concatenated logits.
>
> -----------------
> > Q. What happens if you apply a method like FedAdam to a model whose total size is comparable to the total FENS model
> size?
>
> $\rightarrow$ This is an interesting ablation.
> We evaluated this suggestion by downsizing client local models such that the total size of FENS approximately matches the size of the single model in FedAdam.
> Preliminary results indicate that under a comparable memory footprint, FENS still remains competitive with the original FedAdam.
> On the other hand, using the downsized model for FedAdam induces an accuracy drop.
> We will include the complete ablation in the revised version of our paper.
>
> -----------------
> > Q. How do the results in Figures 2 and 4 change if we instead consider total client computation instead?
>
> $\rightarrow$ There is indeed a trade-off here and we expect the client computation cost of FENS to be the highest in Fig. 2.
> This can be seen as the cost of FENS in providing OFL-like communication efficiency and FL-like accuracy which are its primary metrics of interest.
>
> -----------------
> > On the limitations of synthetically partitioned and vision datasets.
>
> $\rightarrow$ We appreciate the reviewer’s recognition of our use of the FLamby benchmark. We are keen on extending our empirical analysis with a language dataset and will incorporate this in the revised version of the paper.
>
> -----------------
> > On trimming Sections 1 and 2 and including other ablations.
>
> $\rightarrow$ Thank you for the suggestion. We will shrink the first two sections to enable the inclusion of Appendix C and above ablation in the main body of our revised paper.
>
> -----------------
> > On expanding the discussion on privacy.
>
> $\rightarrow$
> Privacy is indeed an important concern and we believe that existing privacy schemes could be directly incorporated into FENS.
> One approach could be to perform client local training using differentially private SGD [1].
> Thus each client can provide a differentially private local model for the ensemble.
> Similarly, the aggregator training could be conducted via a differentially private FL algorithm [2].
> As suggested by the reviewer, we will include a discussion of potential directions in our revised paper.
>
> -----------------
> [1] Abadi, Martin, et al. "Deep learning with differential privacy." Proceedings of the 2016 ACM SIGSAC conference on computer and communications security. 2016.
>
> [2] Noble, Maxence, et al. "Differentially Private Federated Learning on Heterogeneous Data". Proceedings of the 25th International Conference on Artificial Intelligence and Statistics (AISTATS). 2022.

---

### Official Review · Reviewer_3Dng · 2024-07-14

**Soundness:** 3
**Presentation:** 2
**Contribution:** 2
**Rating:** 3
**Confidence:** 4

**Summary:**

This paper focuses on the one-shot federated learning with the model ensembling to assist the model aggregation. Specifically, after only one-round of model uploading, the authors provide a new aggregator based on a shallow neural network for the global model. The performance indicates that the proposed method has good potential in the one-shot FL setups.

**Strengths:**

1. One-shot FL is a very practical scenario in real-world setups, and it has not been richly discussed yet.

2. The usage of the Flamby evaluation datasets is very novel and very close to the real setups.

3. The proposed method is easy to implement and very efficient in communication cost.

**Weaknesses:**

1. The paper lacks several strong baselines in the experiment. Even though the paper focuses on the one-shot FL setups, the design of FENS is very close to the knowledge-distillation-based FL methods, such as FedGKD [1] and Fed-ET [2]. The paper should include them in the evaluation baseline to see whether the proposed FENS could provide SOTA performance. The selected FedKD is a relatively old baseline from which to compare.

2. I suggest expanding the ablation study to discuss the effect of server training in FENS. I noticed that FENS needs to apply global data for training on the server side, which is a very strong setup. I suggest the author elaborate on the impact of the usage of public data.

3. Following my previous point, the usage of public data is sometimes infeasible in the real-world FL setup. In the experiment setup, the author selected proxy data by splitting the original train data. It is a very strong setup. The previous study [2] used the same setups in the paper, but it at least does some distortion to simulate the proxy data coming from a different source. As a result, I feel inconvincible to the
effect of FENS as the server has access to a portion of training data.

[1]. He, Chaoyang et al. “Group Knowledge Transfer: Federated Learning of Large CNNs at the Edge.” arXiv: Learning (2020): n. pag.

[2]. Cho, Yae Jee et al. “Heterogeneous Ensemble Knowledge Transfer for Training Large Models in Federated Learning.” International Joint Conference on Artificial Intelligence (2022).

**Questions:**

1. I am curious about how "Revisiting" is discussed in the paper. I do not see any revisiting part in the writing.

2. I suggest the author to list a pseudo-algorithm in the Section 2 to let the audience better understand the proposed method.

**Limitations:**

Please refer to the Weakness part.

---

> ### Author Rebuttal · Authors · 2024-08-06
>
> We thank the reviewer for the feedback and insightful comments. We noticed that there was a confusion regarding the usage of public dataset in our algorithm. We also noticed that most of the identified weaknesses seem to arise from this confusion. Below, we clarify our algorithm design and address the reviewer's questions.
>
> --------------------
> > On the usage of public dataset at the server.
>
> $\rightarrow$ Our method does not rely on any public dataset.
> In FENS, the data remains decentralized throughout.
> In the second phase of the algorithm, FENS trains the shallow aggregator model via standard FL (lines 131-139).
> We are happy to provide further details or address any additional questions.
>
> --------------------
> > On the lack of strong baselines such as Fed-ET [1] and FedGKT [2].
>
> $\rightarrow$  In light of the above clarification, we do not consider Fed-ET [1] since it is an iterative FL method that relies on the usage of public data.
> Similarly, we do not compare to FedGKT [2] since it is an iterative algorithm that focuses on reducing the computational burden of edge clients.
> Hence, the objectives and design of the suggested baselines differ from that of FENS.
> In particular, FENS focuses on simultaneously achieving OFL-like communication efficiency and FL-like accuracy.
> Accordingly, we have considered 5 one-shot baselines and 6 iterative FL baselines that are more relevant to FENS.
> We will further clarify our choices in the paper.
>
> --------------------
> > Q. I am curious about how "Revisiting" is discussed in the paper. I do not see any revisiting part in the writing.
>
> $\rightarrow$ Ensembling has been studied in OFL prior to our work, and we revisit this concept by applying it in a different and more powerful way.
> Unlike existing methods, FENS employs a shallow aggregator model atop locally trained client models, thereby generalizing traditional aggregation functions.
> By introducing a novel two-phase training procedure, which keeps data decentralized, FENS significantly improves accuracy while maintaining low communication costs.
> We will further clarify the "revisiting" aspect in our paper.
>
> --------------------
> > Q.  I suggest the author to list a pseudo-algorithm in the Section-2 to let the audience better understand the proposed
> method.
>
> $\rightarrow$ Thank you for the suggestion. We will do so in the final version of the paper.
>
> --------------------
>
> [1] Cho, Yae Jee et al. “Heterogeneous Ensemble Knowledge Transfer for Training Large Models in Federated Learning.”
> International Joint Conference on Artificial Intelligence (2022).
>
> [2] He, Chaoyang et al. “Group Knowledge Transfer: Federated Learning of Large CNNs at the Edge.” arXiv: Learning (2020):
> n. pag.

---

> > ### Comment · Reviewer_3Dng · 2024-08-12
> >
> > ```
> > On the lack of strong baselines such as Fed-ET [1] and FedGKT [2].
> > ```
> >
> > I do not think the author's response clarifies my concerns. The baseline selection in the paper is very weak and could not represent the state-of-the-art performance. For the 6 iterative FL baselines, they are actually different aggregation functions. For the one-shot baselines, you select FedKD, which is also a knowledge distillation-based method that is similar to your setups. However, the FedKD is a 2022 paper, so why did the author not select the recent Fed-ET to compare? Fed-ET shares the same setup as FedKD, so why FedKD could be considered as a one-shot FL baseline but Fed-ET could only be considered as an iterative FL method which is out of the topic?

---

> > > ### Author Response · Authors · 2024-08-13
> > >
> > > > Fed-ET shares the same setup as FedKD, so why FedKD could be considered as a one-shot FL baseline but Fed-ET could only be considered as an iterative FL method which is out of the topic?
> > >
> > > $\rightarrow$ As suggested by the reviewer, we have conducted the experiments for Fed-ET and present the results below. Fed-ET, through its advanced aggregation involving weighted consensus and diversity regularization, demonstrates a notable performance improvement over FedENS, which relies on simple averaging. In comparison, Fed-ET and FedKD achieve similar performance, as both employ weighted aggregation.
> > > However, our method FENS still achieves the highest accuracy due to its powerful trainable aggregator stacked atop the ensemble.
> > > We will add these results to our revised paper.
> > >
> > > We hope that these additional results address the reviewer's last remaining concern about our submission. Since we had addressed all other issues in our previous response, we would be thankful if the reviewer could reconsider the score accordingly.
> > >
> > > | Method | α    | FedENS         | FedKD         | Fed-ET        | FENS                    |
> > > |--------|------|----------------|---------------|---------------|-------------------------|
> > > | CF-100 | 0.01 | 16.59 $\pm$ 2.07 | 28.98 $\pm$ 4.55 | 20.37 $\pm$ 1.53 | **44.46** $\pm$ **0.31**  |
> > > |        | 0.05 | 20.56 $\pm$ 3.51 | 39.01 $\pm$ 1.11 | 33.20 $\pm$ 1.34 | **49.70** $\pm$ **0.86**  |
> > > |        | 0.1  | 27.41 $\pm$ 2.71 | 42.38 $\pm$ 0.78 | 38.53 $\pm$ 1.04 | **51.11** $\pm$ **0.37**  |
> > > | CF-10  | 0.01 | 15.66 $\pm$ 6.11 | 18.59 $\pm$ 2.92 | 16.94 $\pm$ 7.88 | **44.20** $\pm$ **3.29**  |
> > > |        | 0.05 | 39.56 $\pm$ 6.33 | 38.84 $\pm$ 6.03 | 37.51 $\pm$ 2.87 | **68.22** $\pm$ **4.19**  |
> > > |        | 0.1  | 48.40 $\pm$ 9.01 | 64.14 $\pm$ 5.17 | 47.06 $\pm$ 2.31 | **75.61** $\pm$ **1.85**  |
> > > | SVHN   | 0.01 | 20.31 $\pm$ 3.49 | 23.62 $\pm$ 10.1 | 12.63 $\pm$ 6.23 | **57.35** $\pm$ **12.6**  |
> > > |        | 0.05 | 38.91 $\pm$ 7.28 | 37.41 $\pm$ 9.62 | 41.14 $\pm$ 6.66 | **76.76** $\pm$ **2.98**  |
> > > |        | 0.1  | 51.99 $\pm$ 7.85 | 61.38 $\pm$ 3.90 | 58.91 $\pm$ 2.81 | **83.64** $\pm$ **0.75**  |

---

### Comment · Area_Chair_psTM · 2024-08-11
**Start discussions with authors, please**

Dear Reviewers,

Thank you for your valuable contributions to the review process. As we enter the discussion phase (August 7-13), I kindly request your active participation in addressing the authors' rebuttals and engaging in constructive dialogue.

Please:

- Carefully read the authors' global rebuttal and individual responses to each review.

- Respond to specific questions or points raised by the authors, especially those requiring further clarification from you.

- Engage in open discussions about the paper's strengths, weaknesses, and potential improvements.

- Be prompt in your responses to facilitate a meaningful exchange within the given timeframe.

- Maintain objectivity and professionalism in all communications.

If you have any concerns or need guidance during this process, please don't hesitate to reach out to me.

Your continued engagement is crucial for ensuring a fair and thorough evaluation process.
Thank you for your dedication to maintaining the high standards of NeurIPS.

Best regards,

Area Chair

---

### Author Response · Authors · 2024-08-14
**Thank you for your engagement and feedback**

As the discussion period is coming to an end, we sincerely thank all the reviewers once again for taking the time to review our work. Your constructive remarks have helped us greatly in improving our paper. We look forward to the opportunity to share our improved work with the community.

---

### Decision · Program_Chairs · 2024-09-25

**Decision:**

Accept (poster)

**Comment:**

**Summary:**

FENS is a novel approach combining one-shot federated learning with iterative training of an aggregator model to improve accuracy while maintaining low communication costs. It uses a two-phase process: 1) clients train models locally and send them to the server, 2) clients collaboratively train a lightweight aggregator model using federated learning. Experiments show FENS can achieve accuracy close to iterative FL methods while requiring much less communication, outperforming other one-shot FL approaches.

**Strengths:**

- Novel combination of one-shot and iterative FL approaches
- Strong empirical results on multiple datasets, including realistic heterogeneous data
- Clear presentation and comprehensive experiments
- Code provided for reproducibility
- Addresses an important practical scenario (cross-silo FL with limited communication)

**Weaknesses:**

- Lack of theoretical analysis/justification
- Privacy concerns with sharing full client models
- High memory requirements on clients
- Limited technical novelty in core method
- Unclear how to optimally set some hyperparameters (e.g. aggregator training iterations)

**Key Discussion Points:**

- *Comparison to state-of-the-art:* Some reviewers felt more comparisons to recent methods like Fed-ET were needed. Authors provided additional results showing FENS still outperforms these.
- *Privacy and scalability:* Concerns were raised about sharing full client models and memory requirements. Authors acknowledged these limitations and suggested future directions like differential privacy.
- *Theoretical grounding:* While lacking formal theory, authors clarified the motivation from stacked generalization literature.
- *Applicability beyond vision tasks:* Reviewers suggested evaluating on language tasks. Authors agreed to add this in revision.
- *Alternative metrics:* Suggestion to c onsider client computation costs in addition to communication. Authors acknowledged this tradeoff.

Overall, most reviewers were positive about the empirical results and practical relevance, while noting some limitations in theoretical grounding and scalability. The authors' responses addressed many concerns, leading some reviewers to increase scores. The consensus seems to be that this is a solid contribution to federated learning research, particularly for cross-silo scenarios with limited communication budgets.